# Snowfall and snow accumulation during the MOSAiC winter and spring season

David N. Wagner[1,2], Matthew D. Shupe[3,4], Christopher Cox[3,4], Ola G. Persson[3,4], Taneil Uttal[3], Markus M. Frey[5], Amélie Kirchgaessner[5], Martin Schneebeli[1], Matthias Jaggi[1], Amy R. Macfarlane[1], Polona Itkin[6,7], Stefanie Arndt[8], Stefan Hendricks[8], Daniela Krampe[8], Marcel Nicolaus[8], Robert Ricker[8], Julia Regnery[8], Nikolai Kolabutin[9], Egor Shimanshuck[9], Marc Oggier[10], Ian Raphael[11], Julienne Stroeve[12,13,14], and Michael Lehning[1,2]

[1]WSL Institute for Snow and Avalanche Research SLF, Davos, Switzerland
[2]CRYOS, School of Architecture, Civil and Environmental Engineering, EPFL, Lausanne, Switzerland
[3]NOAA Physical Science Laboratory, Boulder, CO, USA
[4]Cooperative Institute for the Research in Environmental Sciences, University of Colorado Boulder, Boulder, CO, USA
[5]British Antarctic Survey - Natural Environment Research Council, Cambridge, UK
[6]UiT - The Arctic University of Norway, Tromsø, Norway
[7]Cooperative Institute for Research in the Atmosphere, Colorado State University, Fort Collins, CO, USA
[8]Alfred-Wegener-Institut, Helmholtz-Zentrum für Polar- und Meeresforschung, Bremerhaven, Germany
[9]Arctic and Antarctic Research Institute, St. Petersburg, Russia
[10]University of Alaska Fairbanks, Fairbanks, AK, USA
[11]Thayer School of Engineering, Dartmouth College, Hanover, NH, USA
[12]Centre for Earth Observation Science, University of Manitoba, Winnipeg, Canada
[13]Earth Sciences Department, University College London, London, UK
[14]National Snow and Ice Data Center, University of Colorado, Boulder, CO, USA

**Correspondence:** David N. Wagner (david.wagner@slf.ch)

**Abstract.** Data from the Multidisciplinary drifting Observatory for the Study of Arctic Climate (MOSAiC) expedition allowed us to investigate the temporal dynamics of snowfall, snow accumulation and erosion in great detail for almost the whole accumulation season (November 2019 to May 2020). We computed cumulative snow water equivalent (SWE) over the sea ice based on snow depth and density retrievals from a SnowMicroPen and approximately weekly-measured snow depths along fixed transect paths. We used the derived SWE from the snow cover to compare with precipitation sensors installed during MOSAiC. The data was also compared with ERA5 reanalysis snowfall rates for the drift track. We found an accumulated snow mass of 38 mm SWE between end of October 2019 and end of April 2020. The initial SWE over first-year ice relative to second-year ice increased from 50 to 90 % by end of the investigation period. Further, we found that the Vaisala Present Weather Detector 22, an optical precipitation sensor, and installed on a railing on the top deck of research vessel Polarstern, was least affected by blowing snow and showed good agreements with SWE retrievals along the transect. On the contrary, the OTT Pluvio$^2$ pluviometer and the OTT Parsivel$^2$ laser disdrometer were largely affected by wind and blowing snow, leading to too high measured precipitation rates. These are largely reduced when eliminating drifting snow periods in the comparison. ERA5 reveals a good timing of the snowfall events and good agreement with ground measurements with an overestimation tendency. Retrieved snowfall from the ship-based $K_a$-band ARM Zenith Radar shows good agreements with SWE of the snow cover and

comparable differences as ERA5. Based on the results, we suggest the $K_a$-band radar-derived snowfall as an upper limit and the Present Weather Detector on RV Polarstern as lower limit of a cumulative snowfall range. Based on these findings, we suggest a cumulative snowfall of 72 to to 107 mm, and a precipitation mass loss of the snow cover due to erosion and sublimation as between 47 and 68 %, for the time period between 31 October 2019 and 26 April 2020. Extending this period beyond available snow cover measurements, we suggest a cumulative snowfall of 98 - 114 mm.

# 1  Introduction

Snow cover on sea ice has many significant effects on the ice mass balance and general heat exchange processes between the ocean and the atmosphere (Wever et al., 2020). As snow will cover almost all Arctic sea ice by the beginning of the melt season and with albedo values close to 0.9, a large amount of the incoming solar radiation is reflected rather than absorbed into the snowpack. Due to its potentially very high insulating capacity, snow acts as an inhibitor for heat transfer between ocean,
sea ice, and atmosphere (Holtsmark, 1955; Maykut and Untersteiner, 1971; Sturm et al., 2002b). Depending on the season, accumulation, density, and thermal conductivity of the snow, the sea ice growth and melt varies temporally and spatially. For instance, the underlying sea ice might undergo faster (slower) growth in autumn when the snow on top is relatively thin (relatively thick). On the other hand, a thicker (thinner) snow cover might lead to delayed (earlier) sea-ice melt in the melt season. Consequently, the small-scale snow distribution - which we define in the following as decimeter to hectometer-scale
snow cover area - affects the ice mass balance on the same scales, as large amounts of snow are accumulated along ridges or dunes, while large areas of level ice experience little snow accumulation (Lange and Eicken, 1991; Sturm et al., 1998b; Iacozza and Barber, 1999; Leonard and Maksym, 2011; Trujillo et al., 2016). The snow that has fallen to the ground as fresh precipitation often gets re-distributed as blowing or drifting snow due to the relatively high average horizontal wind velocities during the Arctic winter. The high snow transport rates are also a result of the relatively low aerodynamic roughness length
of sea ice, where $z_0$ is typically lower for first-year ice (FYI) than for second- or multi-year ice (SYI/MYI) (Weiss et al., 2011). In addition, large parts of the snow mass can be expected to get blown into leads or undergo sublimation (Déry and Yau, 2002; Déry and Tremblay, 2004; Leonard and Maksym, 2011; Liston et al., 2020), which has recently been shown to be underestimated by current models (Sigmund et al., 2022). Besides thermodynamic ice growth at its bottom, snow can directly contribute to ice formation on top of the sea ice as snow-ice. Snow-ice formation occurs when snow first transforms into slush
due to surface flooding of saltwater or direct brine expulsion through thin ice followed by subsequent refreezing (Ackley et al., 1990; Sturm et al., 1998a; Toyota et al., 2011; Jutras et al., 2016; Sturm and Massom, 2016). The relative mass contribution of snow-ice towards sea ice by the end of the accumulation season depends strongly on location, with an approximated average of 6 - 10 % for Arctic sea ice, and with estimated local peaks of up to 80 % (Merkouriadi et al., 2020). As a further term in the snow mass balance, Webster et al. (2021) mentions sea ice dynamics. However, we can only imagine that the dynamics, such
as ridge formation, can lead to a snow mass decrease when the snow is pushed below the ice or into the water.
Considering all effects as snow mass source and mass sink, we can write the mass balance equation of snow over sea ice,

modified from the general mass balance description of snow (e.g. King et al. (2008)) as

$$\frac{dM}{dt} = P \pm E_s \pm E_e - E_D - R + B - I - \nabla \cdot D - L - S, \tag{1}$$

where $\frac{dM}{dt}$ is the rate of change of the mass of the snow cover over the sea ice at one point in $\mathrm{kg\,m^{-2}}$, which is equivalent
to Snow Water Equivalent (SWE) per time unit, $P$ is the snowfall rate, $E_s$ is the sublimation rate and $E_e$ is the evaporation
rate of the snow cover, $E_D$ is the drifting and blowing snow particle sublimation rate, $R$ is runoff, $B$ is brine mass infiltration
rate into the snow cover from below, $I$ is the snow-ice formation rate, $D$ is the horizontal snow transport rate of blowing and
drifting snow, $L$ is the rate of the snow mass blown into leads and $S$ is the mass of snow pushed or dug under the ice due to
sea ice dynamics. Considering a larger area (i.e. above hectometer scale up to a scale of the whole Arctic ice pack), all terms
must be considered, while some terms may become zero when considering the equation at one point, e.g. where no open lead
is existent at a point, $L$ becomes zero.

The first and largest source term in Eq. 1 is, depending on the considered area, $P$. The Central Arctic has a dry climate and
depending on location, a yearly average snowfall of approximately 100 to 350 mm can be expected in this area (Serreze and
Hurst, 2000; Chung et al., 2011; McIlhattan et al., 2020; Webster et al., 2021). During polar night, the mass decrease of the
snow cover by sublimation ($E_s$) and evaporation ($E_e$) can probably be assumed as negligible as well as the mass increase due
to deposition (re-sublimation) and condensation (Liston et al., 2020; Webster et al., 2021). However, sublimation and evap-
oration terms become larger by beginning of summer in May and stay relatively large until September. Reliable values from
literature are hard to determine, but the snow cover decrease as combination of $E_s$ and $D$ (as snow particles that get lifted
into suspension) may be up to 50 % (Essery et al., 1999). To estimate the blowing snow sublimation $E_D$, Chung et al. (2011)
applied the often and in various forms used sophisticated PIEKTUK blowing snow model (e.g. Déry et al. (1998); Déry and
Yau (1999, 2002); Déry and Tremblay (2004); Leonard et al. (2008); Leonard and Maksym (2011)) for a SHEBA (Surface Heat
Budget of the Arctic Ocean) field experiment (Uttal et al., 2002) site, drifting between 74 $^\circ$ and 81 $^\circ$ N. They computed 12 mm
of SWE blowing snow sublimation over a time period of 324 days between November 1997 to September 1998. As 179 mm
of precipitation was found for the same time period, the blowing snow sublimation mass sink was 6 % of the total cumulative
snowfall. During the Canadian Arctic Shelf Exchange Study (CASES) overwintering campaign, Savelyev et al. (2006) found
most of the time a relative humidity of over 95 % and concluded on very low blowing snow sublimation rates. Liston et al.
(2020) however, suggested based on modelling results a significant mass reduction of the snow cover by 20 % due to blowing
snow sublimation. Within the melt season in summer, $R$ can be expected to be the largest mass sink (Webster et al., 2021).
Considering brine infiltration, $B$, which is often accompanied by the expression of frost flowers, Nghiem et al. (1997) found in
indoor experiments a 4 mm slush layer forming beneath frost flowers. However, when snow falls onto frost flowers or a layer
of brine, it gets soaked by brine, transformed into slush and, when cold enough, it is often transformed quickly into snow-ice.
Hence, we assume that brine only can be a positive mass term as long as a certain ambient temperature is not undercut, where
the snow begins to transform into snow-ice. Regarding mass decrease due to snow-ice formation $I$, Merkouriadi et al. (2020)
gives an average value of less than 0.05 m snow-ice thickness for the Central Arctic. It is hard to estimate from the 0.05 m a

precipitated amount of SWE from recalculation, as the process of snow-ice formation is complex (Jutras et al., 2016). However, when we assume 0.05 m as snow height with an average fresh snow density of $100\,\mathrm{kg\,m^{-3}}$, we expect around 5 mm of SWE decrease, which would mean only about 3 %, relative to the measured 179 mm during SHEBA. The snow-ice formation rate is expected to be highest in the months of September, October and November (Webster et al., 2021). On one hand, the largest sink term in Eq.1 is the erosion outside the melting season, represented as $D$ in the mass balance equation, which may make up to 50 % SWE decrease over sea ice of the total precipitated snow mass (Leonard and Maksym, 2011). On the other hand, locally, the eroded mass may deposit at the windward and leeward side of ridges, on level areas as dunes and fills frozen leads - hence locally very often exceeds the precipitated mass. The amount of drifting and blowing snow that is lost and gets melted in open leads $L$ varies strongly depending on location, considered area, ice dynamics and lead properties such as width and orientation relative to the wind. However, the total vanished mass flux from the column of blowing and drifting snow can make locally up to 100 % (Déry and Tremblay, 2004; Leonard and Maksym, 2011). Déry and Tremblay (2004) computed for a 10 km fetch, using the blowing snow model PIEKTUK with a mean lead width of 100 m, an open water fraction of 1 % and a typical lead trap efficiency of 80 %, an annual blowing snow loss of 20 mm SWE. However, in this model setup, saltation mass flux is not considered. Leonard et al. (2008); Leonard and Maksym (2011) were doing computations with the same model-base for Antarctic sea ice, but considering saltation mass flux in addition. They emphasize the relative importance of saltation mass flux in the computation, as they find that all saltated mass flux blown towards an open lead vanishes there, and that although the mass flux within the saltation layer in their model is lower than in the blowing snow column above, the higher frequency of saltation (about 50 % on 23 days in October 2007) compared against blowing snow frequency makes the mass loss due to saltation an important term. However, only a very limited number of studies was carried out that investigate this specific problem and the saltation layer with relative large snow mass flux was not considered in great detail so far. Hence, the existing estimates go along with large uncertainties.

As we will only consider the accumulation time period, we can omit runoff $R$ from Eq.1. Further, snow cover evaporation and sublimation terms are negligable during this time, hence we can neglect the terms $E_s$ and $E_e$. Then we write the simplified mass balance equation for winter and early spring as:

$$\frac{dM}{dt} = P \pm D - L - E_D - I - S. \tag{2}$$

To investigate all effects of the snow cover over the ice - the insulating effect, the sea ice mass-contribution effect, and the albedo effect - light must be shed into the snow processes that are represented and detailed knowledge of the evolution of total snow mass $\frac{dM}{dt}$, or SWE over time on top of the ice, is required. However, due to logistical challenges, especially for the winter and spring months, snowfall rate and snow accumulation estimates could only be roughly approximated so far. The past estimates mostly made use of rare point measurements, or rather old time series (Petty et al., 2018) and satellite remote sensing (Petty et al., 2018; Cabaj et al., 2020), leading to high uncertainties in weather-, climate and snow cover models as well as in reanalyses. Batrak and Müller (2019), for instance, could show that a 5 to 10 °C warm bias of the sea ice surface temperature in weather forecasts and reanalyses is due to a missing snow layer modeled on top of the sea ice. For snowfall

rates and mass balance estimations, some general problems occur: Limited data about snowfall rates from precipitation gauges currently exist for this region. Buoys that measure snow height with acoustic sensors which record long continuous time series along its drift tracks throughout the Central Arctic do exist (Nicolaus et al., 2021a). However, uncertainties with point snow measurements arise in those windy regions due to the snow transport processes described above. If using precipitation sensors, the high average horizontal wind velocities make snowfall rate estimates difficult for both weighing gauges (Goodison et al.,
1998) and optical sensors (Wong, 2012a). The wind itself may lead to an undercatch for weighing bucket gauges (Goodison et al., 1998), while blowing snow may lead to overestimation for both, weighing gauges and optical sensors (Sugiura et al., 2003). Blowing snow typically occurs at heights up to ten meters, while it can even reach several hundreds of meters in altitude (Budd et al., 1966; Scarchilli et al., 2009). Hence, we expect that blowing snow can often falsely be detected as precipitation by snowfall sensors (Sugiura et al., 2003). Some issues caused by the wind can be corrected with scaling factors or transfer
functions, but these need to be identified for these specific conditions (Goodison et al., 1998). Another approach is to measure the snow water equivalent (SWE) of the snow cover. From this, one can derive snowfall rates. However, especially during the polar night, the precipitated snow is dry, and as already indicated above, the wind speed is often sufficiently high to drift the freshly fallen snow particles away immediately. Hence, single point measurements are not appropriate to estimate snowfall and horizontal sampling distance and temporal distance between sampling days should be kept as short as possible. This becomes
more crucial the more windy the location is.

During the year-long Multidisciplinary drifting Observatory for the Study of Arctic Climate (MOSAiC) expedition, during which the research vessel (RV) Polarstern (Alfred-Wegener-Institut Helmholtz-Zentrum für Polar- und Meeresforschung, 2017) served as a base moored on two different ice floes, data of snow on the ice as well as of in-situ snowfall was collected
in great detail for almost the whole MOSAiC period (October 2019 - October 2020) (Nicolaus et al., 2021b; Shupe et al., 2022). The dataset includes measurements of the penetration force into the snowpack with a SnowMicroPen (SMP) (Schneebeli and Johnson, 1998; Schneebeli et al., 1999) from which snowpack densities can be estimated (Proksch et al., 2015), bulk SWE measurements, weekly repeated transects of snow depth measurements, and a set of precipitation sensors installed on the ice (Vaisala Present Weather Detector 22 (PWD22) (Vaisala, 2004; Kyrouac and Holdridge, 2019), OTT Pluvio[2] pluviometer
(Bartholomew, 2020a; Wang et al., 2019b), OTT Parsivel[2] (Bartholomew, 2020b; Shi, 2019), and on board RV Polarstern (Vaisala PWD22, OTT Parsivel[2]).

This paper investigates the snow accumulation period from October 2019 to May 2020, where precipitation is solid, and no significant snowmelt was observed. For this period, the intentions in this paper are as follows:

– Compute reliable values for SWE evolution along the fixed transect paths that include surface heterogeneities

   – Use the computed SWE for periods where no drifting snow occurred to compare with snowfall rates from precipitation gauges installed during the MOSAiC expedition and make a best-estimate of total precipitation during the investigation period

- Evaluate an existing radar reflectivity - snowfall ($Z_e$-S) relationship (Matrosov, 2007; Matrosov et al., 2008) for the ship-based $K_a$-Band ARM Zenith Radar (KAZR)

- Evaluate the ERA5 (Hersbach et al., 2020) mean snowfall rates for the MOSAiC drift track

- Investigate average snow mass balance and discrepancies of computed snowfall rates and SWE on the sea ice and shed light into the processes described in Eq.2, such as total eroded mass

Section 2 introduces our methods, followed by section 3, where we show the results. In section 4 we discuss our results, and in section 5 we draw conclusions about our findings and give an outlook about potential future work.

## 2  Data and methodology

All data used for evaluations in the following was collected during the MOSAiC campaign (Krumpen et al., 2020; Nicolaus et al., 2021b; Shupe et al., 2022) from the beginning of Leg 1 (24 October 2019) until the end of Leg 3 (7 May 2020) (Fig. 1). On 4 October 2019, RV Polarstern moored along an ice floe that originated in the Siberian shelf (Krumpen et al., 2020).

### 2.1  Ice conditions and Central Observatory

According to Krumpen et al. (2020), the floe where RV Polarstern moored on had a size of approximately 2.8 km x 3.8 km and was a loose assembly of pack ice, little less than a year old that had survived the 2019 summer melt. Fig. 2 shows a map of the ice- and snow surface structures and installations by 5 March 2020 of the MOSAiC Central Observatory. Note that the shown elevation range is only approximate as problems occurred with the inertial navigation system of the laser scanner. This lead to tilts and the single swaths within the map have staggered heights. To this date, these uncertainties could not be corrected. However, very bright areas indicate ridges of around 2 m height, with locally 3 m height and more. The Central Observatory (all installations in the close vicinity of Polarstern) was distinguished from the Distributed Network, which consisted of remote autonomous stations at least a few kilometers away from the CO. The detailed concept about the Central Observatory and Distributed Network are explained in Nicolaus et al. (2021b). The core of the floe consisted mostly of deformed second-year ice (SYI), and the ice surrounding this core mainly consisted of frozen melt ponds (remnant SYI) and partially first-year ice (FYI). When the ship moored, the heading of RV Polarstern was about $220\,^{\circ}$ in October 2019. Significant changes in ice conditions occurred the first time around 16 November 2019, when a storm led to strong ice deformations in and around the CO. Another significant ice deformation event occurred around 11/12 March 2020 and other periodically until 7 May 2020. Over time, the floe rotated counter-clockwise and reached a minimum heading of $75\,^{\circ}$ on 21 March 2020.

We describe the measuring setup and the post-processing for all used data streams in the following.

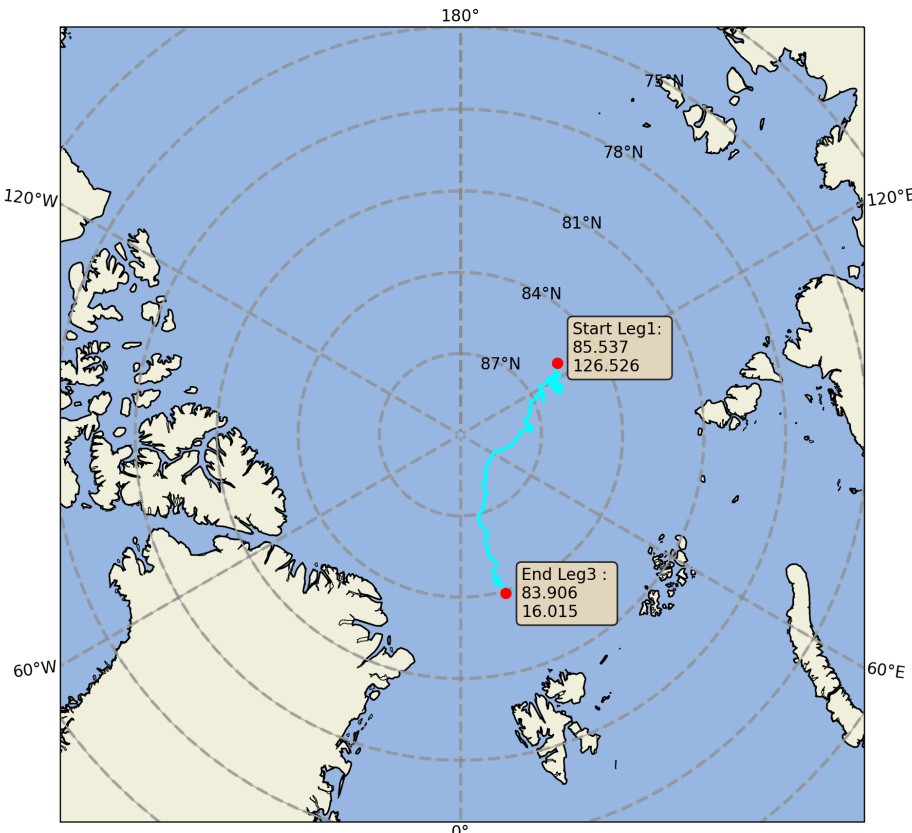

**Figure 1.** The drift trajectory of RV Polarstern between 24 October 2019 and 7 May 2020.

## 2.2 Snow cover measurements

### 2.2.1 SMP force and SWE measurements

We measured snow-water equivalent (SWE) with an ETH tube, a SWE sampler that is commonly used in Switzerland (Haberkorn, 2019; López-Moreno et al., 2020), as well as resistance force with the SnowMicroPen (SMP) (Schneebeli and Johnson, 1998; Schneebeli et al., 1999) and snow height at different sites (areas shaded in yellow in Fig. 2). The bulk SWE measurements with the ETH cylinder follow the simple principle where the mass of the snow fitting in a tube with a known cross-sectional area is weighed on a spring scale, which yields the SWE in mm, or $kg\,m^{-2}$. The device is calibrated for low temperatures, which is most important for the steel spring of the scale. López-Moreno et al. (2020) made an intercomparison of various bulk density and SWE samples including the ETH tube and tested for instrumental bias and variability. It can be concluded that from a single ETH tube measurement, we might expect a maximum error of 10 %. This value appears high, but given the fact that the average Arctic sea ice snow cover is thin, the absolute error will be low. Given a 20 cm snow depth, the maximum expected error would only be 2 cm. López-Moreno et al. (2020) also argue, that particularly light samples may lead to an additional

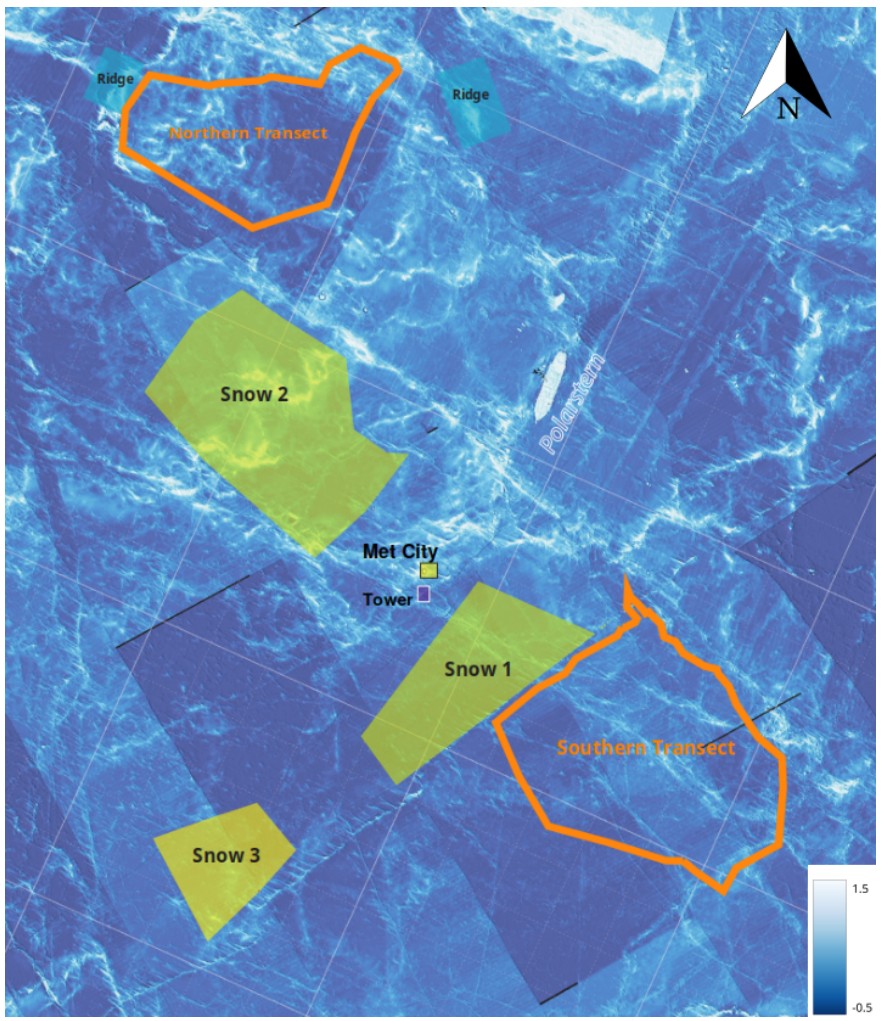

**Figure 2.** Main snow measuring areas on the MOSAiC floe by 5 March 2020. The bottom layer is a digital elevation map (DEM) from airborne laser scanning (ALS) with the helicopter. The square side length of the underlay grid is 500 m. The transect paths and margins of the shaded measuring areas are based on GPS measurements. The legend for elevation is shown in meters. However, the elevation range is only approximate due to issues with the inertial navigation system which could not be corrected so far.

10 % of error with respect to the weighing process with the spring scale itself. Nonetheless, a currently non-quantified error is that during a bulk SWE measurement a sharp transition between snow cover and sea ice often cannot be determined, which is especially valid for an underlying surface scattering layer (SSL) on SYI. However, we use a relatively big sample size of $n = 195$ bulk SWE measurements, and with increasing sample size, uncertainties are expected to be increasingly leveled out. To avoid wind influence on the measurements, the weighing was each conducted in the wind shadow of surrounding objects, persons or the person measuring itself.

The SMP is a device which measures the penetration resistance force (N) by means of a rod with a conic tip that is slowly driven vertically into the snowpack. A force sensor is connected to the tip which detects the force that is needed to drive into the snowpack with $\mu$m resolution. The output is given as a force-snow depth signal. These penetration resistance force signals can be used to estimate snowpack density and detect the layers in the snowpack (Proksch et al., 2015; King et al., 2020). We used three different sensors, but all SMP version 4, during MOSAiC. Processing of density from SMP force signals is discussed in the next chapter.

The map in Fig. 2 shows the floe state on 5 March 2020, which changed significantly due to ice dynamics that started on 11 March 2020. Snow was measured at the different sites as well as along both transect loops. The measurements cover a large area of the floe, including level, remnant SYI, FYI, and deformed SYI. Details about the sampling procedure will follow below. Snow 1 was characterized mainly by a mixture of remnant SYI and deformed SYI. In the beginning, Snow 1 was mostly flat, but the surface became rougher over the time of the expedition. At Snow 1 we deployed three snow pit sites, which were maintained until the end of leg 3. The Snow 2 plot was characterized as an open level field, mostly on remnant and deformed SYI with a distinct, high, and long pressure ridge in the center of the plot. On Snow 2, we maintained two snowpit locations until the end of leg 3. Both sites had very similar underlying ice conditions. Snow 3 was created at a later point, furthest away from the vessel. In the beginning, it was a very flat area with underlying FYI and was maintained during leg 3 but needed to be abandoned due to ice dynamics in mid-March. Further, weekly snow pit measurements were conducted along the south-westerly section of the northern transect loop. Also, transects were conducted infrequently over ridges, and measurements were conducted weekly at the ice coring sites during Leg 1 (beyond the map boundaries in Fig. 2, but located north-west of the ship), among other measuring locations. The large variety of locations, their underlying ice types, and snow depths allow us to take the spatial heterogeneities of the snow cover into account. However, since we use a bulk approach with the collected SMP and direct SWE data, detailed information on each measuring site is not needed and will not be provided here.

At the measuring locations, SWE, snow height, and penetration resistance force measurements with the SMP were done. The SMP measurements at the recurring snowpit locations were conducted as follows: Five SMP measurements were performed at a distance of about 20 cm along a line parallel to the old snowpit wall to account for the spatial heterogeneity of the snowpack. On the ridge sites, for instance, SMP measurements were conducted infrequently as transects over ridges. We used these measurements to estimate SWE along the northern and southern transect loops, which will be explained below. For more details

about the SMP and SWE collection, we refer to Macfarlane et al. (2022a, b); Wagner et al. (2021) (data publicly accessible after end of the MOSAiC moratorium in January 2023) and soon to be published data and method papers by MOSAiC participants that describe the MOSAiC snow measurement setup in detail.

### 2.2.2 Transects

Snow depth transects were conducted weekly with a Magnaprobe (Sturm and Holmgren, 2018; Itkin et al., 2021), if the atmospheric, ice, or overall safety conditions did not prevent it. The transect path was distributed into two loops (Fig. 2): A northern loop, mostly situated on deformed SYI, and a southern loop, which was mainly situated on FYI and remnant SYI with underlying frozen melt ponds. A transition zone distinguished these two loops, mostly consisting of frozen melt ponds with a very flat surface without significant heterogeneities. The approximate ice conditions and the transect loop locations can be seen in orange on the MOSAiC floe map from 5 March 2020 (Fig. 2). The elevations on the southern transect are mostly below around 1 m height. The elevations are generally higher on the northern transect, although it does not cover ridges of up to 3 m height or more as they have been observed on the Snow 2 plot. One important aspect to note is that surrounding elevations of the ice (i.e. ridges) often exceed the height of the transect areas, for instance in the north and the south of the northern transect, or in the north and the east of the southern transect. The expected result from this surrounding sea ice characteristic is that during drifting snow events, snow might get caught upwind of the transect areas, while wind speeds may decrease, potentially leading to flow separation and a bias in the total snow accumulation. We consider this as a non-quantifiable uncertainty at this point of research. The only way to overcome this problem in the future is to cover larger sampling areas and increase the number of these. To find an optimum between technical effort and reliable result, one could decrease these areas until average values are not changing any more significantly.

The GPS coordinates of the Magnaprobe were transformed into coordinates of a local metric coordinate system, called "FloeNavi" (Fig. 3). Furthermore, the transects were partially corrected for shifts within the ice, which was especially the case for an event with strong ice dynamics on 16 November 2019. Thus, the southern loop transects until (including) 14 November 2019 are marked as a yellow rectangle in Fig. 3. Note that a part of the northern transect was off the regular transect track on 31 October 2019, which leads to some uncertainties in evaluations. However, we tested how it affected the average when only the off track was cut off versus the whole northern transect. We found an increase of only 0.8 mm when the whole track was considered. Hence, though this leads to some uncertainties, we included the 31 October 2019 transect for further evaluations. For all other days of sampling, though the transects may deviate from one another within the FloeNavi coordinate system, the actual transect path was the same. After the coordinate transformation and horizontal correction, for good spatial and temporal comparability, clear margins as shown in the rectangles in Fig. 3 were defined for the "southern" and "northern" transect loop. By the overlays, one can recognize that the transect loops were not significantly impacted by internal differential ice movements.

However, ice dynamics affected the transects especially from 11/12 March 2020 on, where leads and cracks opened throughout the paths. Overall, we tried to minimize the influence of these ice deformation events on the transect measurements. However, an impact on the time series cannot be excluded. On the transects, snow height measurements were sampled with the Magnaprobe with an average distance between measuring points of 1.1 m. Note that this value is simply an average that contains the uncertainty of GPS localization, coordinate transformation, and the step length of the user, while the users varied mainly between each leg of MOSAiC. The average distance between measurements was computed after applying the FloeNavi coordinate transformation. Values $z < 0.00\,m$ as well as $z > 1.40\,m$ (the technically constrained measurable length) were discarded as incorrect data. In this study, we did not account for further corrections that may come along with a tip sinking into material below the snow cover, for instance the surface scattering layer on SYI. Sturm and Holmgren (2018) showed that this error is hard to quantify, as it depends strongly on the ground material and of course the applied force as well - an issue which also occurs for crusts within the snow cover. Similar issues occur for other snow measurements, such as ETH tube and SMP, as well, which will be mentioned later in the text again.

The northern loop was sampled from 24 October 2019 to 7 May 2020 on 24 days with an average path length of 954 m. The southern loop was sampled from 31 October 2019 to 26 April 2020 14 times with an average transect path length of 974 m. More in-detail data and instrument description is found in Itkin et al. (2021).

To take surface roughness (i.e. variability of snow surface height) and potential snow accumulation at surface irregularities better into account, we looked at the weekly snow height differences of the transects. With the given average horizontal sampling distance (1.03 - 1.21 m), no small-scale patterns are considered (sastrugi, for instance) for evaluation. However, since the extent of ridges and most types of dunes are in all horizontal directions larger than 1.2 m (Filhol and Sturm, 2015), we expect our typical horizontal sampling scale to accurately characterize the spatial distribution of accumulation, which we demonstrate in Sec. 2.3.1.

## 2.3 SMP density retrievals and SWE from transect snow depths

Based on how the campaign was planned, we have considerably more SMP force measurements available ($N = 3007$) than bulk SWE weighing measurements ($N = 195$). Furthermore, for each snowpit we made at least $n = 5$ SMP measurements and the SMP was often used even for ridge transects, these measurements best characterize the spatial heterogeneity in the snow depth across the sea ice. Not many direct SWE measurements or SMP force measurements are available along the transect path. Hence, we use the direct SWE measurements for validation but apply a statistical SWE - snow depth (HS) relationship to estimate SWE along the full path (Sturm et al., 2002a; Jonas et al., 2009).

Snowpack density can be estimated with a statistical model from SMP snow depth-force signal profiles (Proksch et al., 2015):

$$\rho = a + b \ln\left(\tilde{F}\right) + c \ln\left(\tilde{F}\right) L + dL \tag{3}$$

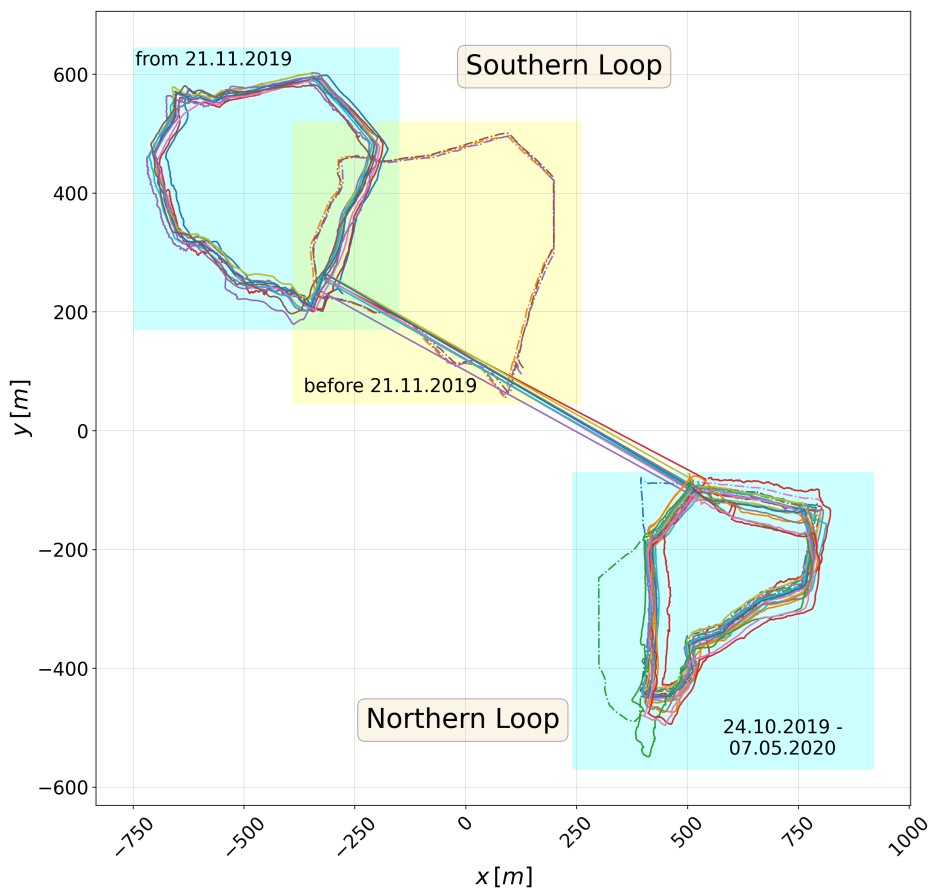

**Figure 3.** Magnaprobe transect paths with coordinates transformed to the FloeNavi grid corrected for ice drift. The rectangles represent the margins that were used as a definition for "Northern transect loop" (upper left) and "Southern transect loop" (bottom right) for good comparability. The shifts between transect paths within a rectangle originate from corrections and coordinate transformation, though the actual transect paths were the same.

where $a\left(kg\,m^{-3}\right)$, $b\left(N^{-1}\right)$, $c\left(N^{-1}\,mm^{-1}\right)$, $d\left(mm^{-1}\right)$ are empirical regression coefficients, $\tilde{F}$ is the median penetration force of the SMP ($N$) for a specified sliding window and $L$ is the microstructural length scale (Löwe and Herwijnen, 2012) for the same window. Both $\tilde{F}$ and $L$ are computed for the window size of 2.5 mm with a 50 % overlap, which is the same as used in Proksch et al. (2015), but contrary to Calonne et al. (2020) (1 mm) and King et al. (2020) (5 mm). King et al. (2020) calibrated the corresponding coefficients to snow on Arctic sea ice and found $a = 315.61\,kg\,m^{-3}$, $b = 46.94\,N^{-1}$, $c = -43.94\,N^{-1}\,mm^{-1}$, $d = -88.15\,mm^{-1}$. The coefficients show a significant improvement in density derivation for snow on sea ice, which is reflected by the decrease in the root-mean-square error (RMSE) (Proksch et al. (2015): $RMSE = 130\,kg\,m^{-3}$, King et al. (2020): $RMSE = 41\,kg\,m^{-3}$) without removing outliers, compared against density cutter measurements. Consequently, we used the coefficients from King et al. (2020) for the following SWE computations. From the SMP density estimates

we can compute

$$SWE = HS \cdot \overline{\rho} \tag{4}$$

where $HS\,(m)$ is the height of snow over the ice or snow depth, and $\overline{\rho}\,\left(kg\,m^{-3}\right)$ is the vertically averaged density of the snowpack. The computed SWE dataset is documented in detail by Wagner et al. (2021). Similar to Jonas et al. (2009), but applying the function directly to SWE, we fitted the following function to the available bulk SWE measurements as well as SWE retrievals from the SMP:

$$SWE = m \cdot HS^a \tag{5}$$

where $m$ is the fitted slope and $a$ is a fitting coefficient. For the bulk SWE measurements, we found $m = 334.61$ and $a = 1.14$ (Fig. 4). For the SMP retrievals we found $m = 323.97$ and $a = 1.07$. As we have more SMP measurements available in total, and especially for deep snow depths compared with ETH tube, we computed SWE based on fitted SMP density - SWE parameters as:

$$SWE = 323.97 \cdot HS^{1.07}. \tag{6}$$

From Fig. 4 one can clearly see that the improvement for snow on sea ice of the coefficients found by King et al. (2020) is valid for MOSAiC leg 1-3 SMP data, too (Fig. 4c) and that the coefficients determined by Proksch et al. (2015) and Calonne et al. (2020) appear not appropriate to estimate SWE of snow for this MOSAiC period. Furthermore, the lowest RMSE (expressed here as average error of individually computed SWE relative to the regression line for an individual parameter setup) was found for the fitted model with the coefficients from King et al. (2020) (7.2 mm SWE) compared against 15.4 mm (Proksch

et al., 2015) and 9.4 mm (Calonne et al., 2020). However, one should note the following limitations in this comparison: First, we used for all computations a sliding window size of 2.5 mm, which is the same as in Proksch et al. (2015), while Calonne et al. (2020) used 1 mm and King et al. (2020) a 5 mm window. However, the strength of the influence can at least partially invalidated by the fact that Calonne et al. (2020) state that they tested for sensitivity of three different window sizes of 1, 2.5 and 5 mm and could not find a significant influence on the result - which is not quantified in the publication. At least choosing a

fixed window for each parameterization - as we done with the 2.5 mm window - increases the comparability. Another limitation might be that the Proksch et al. (2015) calibration was made with a SMP version 2, while Calonne et al. (2020), King et al. (2020) and we use the newest SMP version 4.

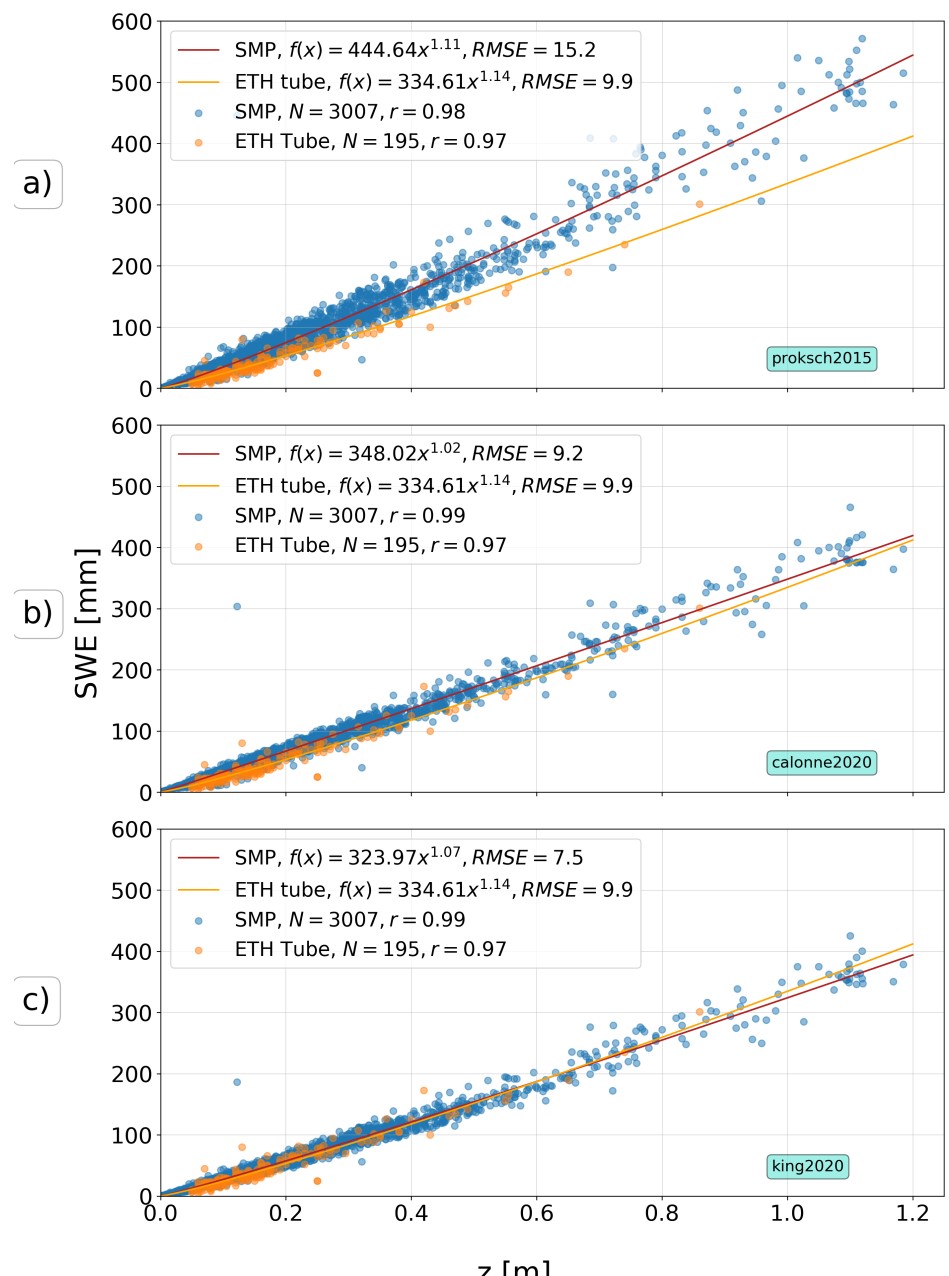

**Figure 4.** Scatter plots and fitted HS-SWE function of SMP derived SWE and measured SWE with ETH tube for a) The density computation coefficents from Proksch et al. (2015), b) Calonne et al. (2020) and c) King et al. (2020).

We applied our fitted formula to each snow depth measurement with the Magnaprobe along the transect path to obtain the SWE estimates. The SWE was rounded to integers for the following description in the text, except when two values that are

compared are very similar. No rounding was conducted before any computations. The average computed SWE will also serve as reference comparison with snowfall sensors and ERA5 as described in section 2.8.

A limitation with this approach is that different snow layers are not distinguished by density, even though a wind-packed layer has a higher density than a depth hoar layer. Hence, when high winds lead to drifting snow deposition that is detected by a snow height increase with the Magnaprobe, the SWE increase is likely to be underestimated, as would be the eroded mass

of a drifting snow layer. It is beyond the scope of this study to attempt an approach that distinguishes different snow layers. Instead, for validation, we compared SMP and ETH tube measurements with transect-computed SWE along a section of the northern transect loop for 14, 21, and 28 November 2019 (Fig. 5). The validation measurements were conducted at different positions at each day of measurements and contain drift locations and level ice areas. Note that this quantitative comparison of bulk SWE/SMP SWE versus transect contains uncertainties as the accuracy of GPS measurements (2 m) and the following

coordinate transformation of the Magnaprobe as well as the SMP coordinates do not allow for cm-scale precision. Further, the pits were dug up to a vertical distance of 1 m from the transect path, in order to sample fresh snow that is not disturbed by repeated transects. For quantitative comparison, SWE computations from direct bulk SWE and SMP measurements along the transects were plotted over SWE model retrievals. Fig. 5a shows the measuring locations for each SMP measurement along the northern transect loop (5 measurements at each snowpit location) and Fig. 5b, Fig. 5c and Fig. 5d show the corresponding

SWE plotted over the x-axis of the FloeNavi for different days of measurements.

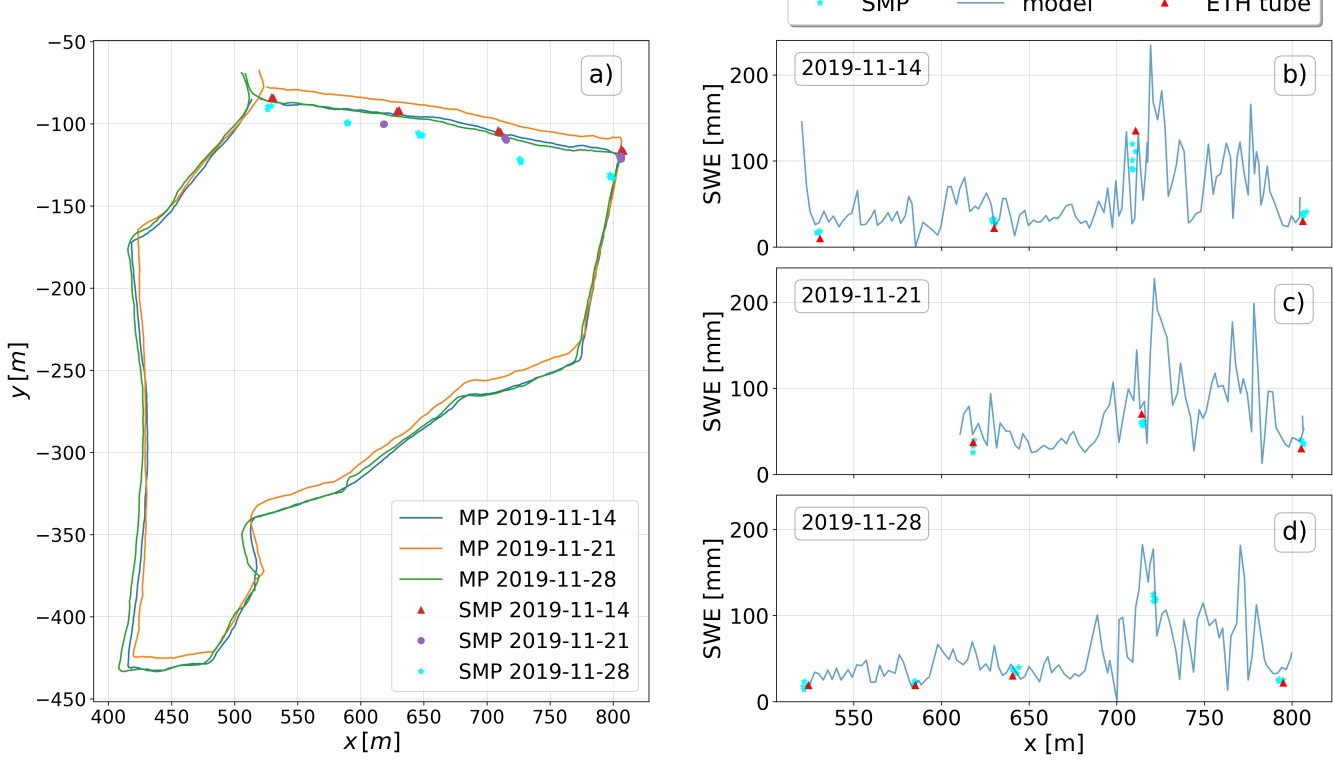

**Figure 5.** a) SMP measurement locations along the Magnaprobe (MP) transect path on 14 (2019-11-14), 21 (2019-11-21), and 28 November 2019 (2019-11-28). The GPS coordinates were transformed into local FloeNavi grid coordinates. b), c), and d) show the comparison of SWE estimates from direct SMP measurements, direct bulk SWE measurements, and SWE derived with the HS-SWE model from the Magnaprobe snow depth measurements along the northern transect as x-axis location on the FloeNavi grid, for 14 , 21 and 28 November 2019.

This comparison shows that the modeled SWE matches the derived SWE from SMP retrievals and bulk SWE measurements quite well during the three chosen time periods, even for higher SWE estimates, where a higher scatter is expected (Fig. 4). Note that although the time distance between the three days of measurements is relatively short, we found from and including 14 November until and including 21 November 2019 on 42 % of the time drifting snow conditions, where the threshold friction velocity for snow transport was exceeded (the lower Snow Particle Counter (SPC) was not installed yet at this time). From and including 21 November until and including 28 November 2019 the threshold was exceeded on 57 % of the time. This means we can expect re-distributed snow for the two following days after 14 November. Inter-compared SMP-SWE versus ETH-SWE, we find a RMSE of 16.3 mm for 14 November, 8.1 mm for 21 November and 3.5 mm for 28 November. However, we must note that the depth of SWE measurements from ETH tube and SMP have some individual but differing restrictions: Firstly, as the SMP cut-off force signal was set to 40 - 41 N (depending on the device), the snow depth was determined whenever one of those values was reached, which is not necessarily the snow - ice interface. Secondly, during the sampling period, there was no method established to distinguish between surface scattering layer (SSL) and snow. Hence, its vertical position was determined

visually, which was not always clear. Therefore, a measurement with the ETH tube might or might not include the surface scattering layer which formed during the melt season of 2019. If the SMP was able to penetrate the SSL only partially while it was not measured with the ETH tube, then SWE is overestimated from the SMP measurements. Otherwise, if the SMP could not penetrate the SSL while it was partially measured with the ETH tube, the SMP-based SWE computation overestimates actual SWE. However, as the SMP-SWE retrievals are often close to the direct SWE measurements, one can assume on the whole reliable values. Research to determine exact boundaries between snow and sea ice is ongoing. Furthermore, since the number of measurement points along a transect is large and we do not expect systematic biases, we believe that fluctuations caused by these various sources of uncertainty will largely average out, such that the results from the applied SWE model yield a reasonable estimate along the transect.

Under section A we make a comparison with derived SWE over Arctic sea ice during the SHEBA campaign conducted by Sturm et al. (2002a) in a similar manner. The comparison shows the difference of their and our results and underlines the importance of using our approach for MOSAiC snow cover data.

### 2.3.1  Evaluating the sensitivity of the arithmetic mean with respect to horizontal sampling distance

As shown by Trujillo and Lehning (2015), a sufficiently small sampling interval of point measurements is crucial for estimating representative values of spatially averaged snow depths. We studied the sensitivity of the horizontal sampling interval for average mass estimates by reducing the sample numbers from using all samples (Magnaprobe average sampling distance of 1.1 m) down to considering every 10th sample (about 11 m sample distance) for the average. The process was conducted for each day of sampling and the averages were normalized against the original sampling frequency (Fig. 6).

The results show that for sampling frequencies down to 1/3 of the original frequency (sampling distances ranging from 1.1 to 3.4 m), the average mass estimates vary by less than $\pm 1 \%$. This indicates that a sampling distance up to 3.4 m is mostly robust and that no significant undersampling occurred. This also shows that the impact of variations in sampling interval distance that inevitably occurs with different operators of the Magnaprobe is probably negligible. The larger fluctuations in computed average mass for longer sample interval distances suggests undersampling at those scales and less reliable averages. However, a validation of uncertainties that could accompany varying vertical penetration force leading to different measured snow height, e.g. when a crust within the snow is penetrated or not due to varying operators, is not conducted here. The operators were aware of this issue and tried to apply a similar power for the Magnaprobe sampling.

### 2.4  Snowfall rates

### 2.4.1  Precipitation gauges

Snowfall rates were estimated using standard internal processing software from five distinct precipitation gauges operated by the US Department of Energy Atmospheric Radiation Measurement (ARM) Program. Two sensors investigated here were installed on the railing on the top deck of Polarstern - a Vaisala Present Weather Detector 22 (Vaisala, 2004; Kyrouac and Holdridge, 2019) (in the following referred to as PWD22PS) and an OTT Parsivel[2] laser disdrometer (Shi, 2019; Bartholomew,

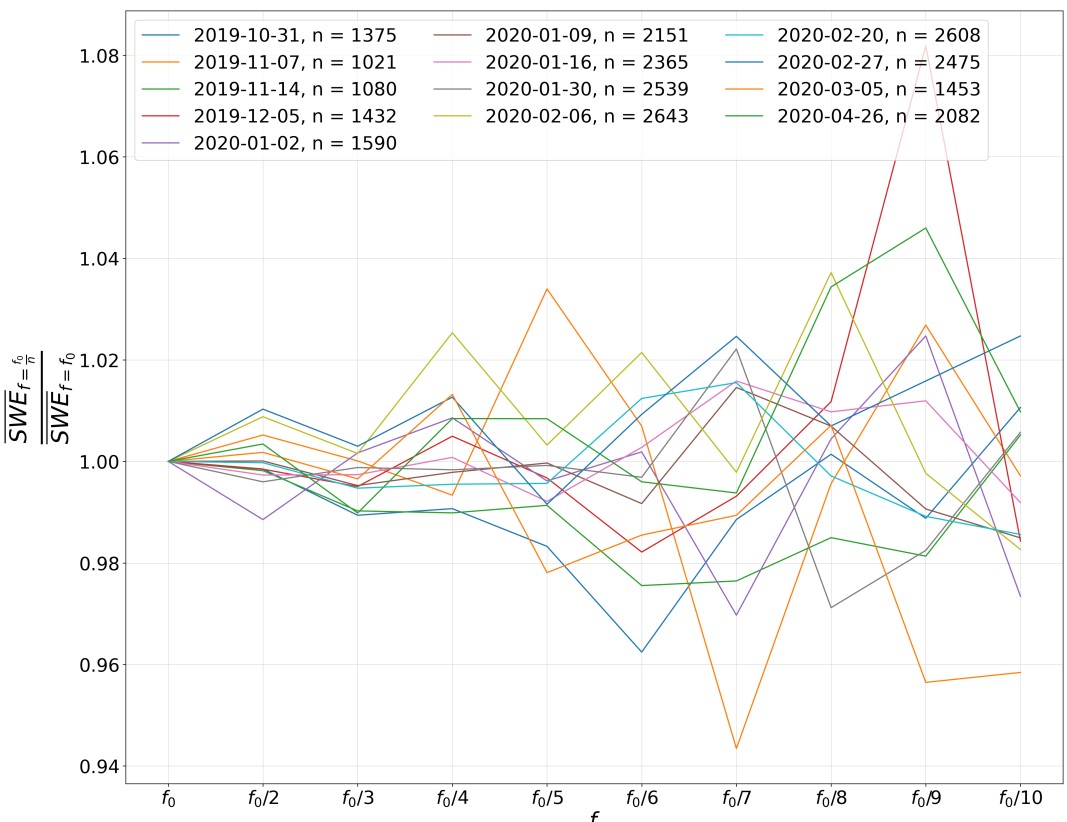

**Figure 6.** Sensitivity of the transect average in dependence of the Magnaprobe horizontal sampling. The x-axis shows the horizontal transect sampling frequency in relation to the original sampling frequency $f_0$, the y-axis shows the ratio of the average SWE of all tested frequencies to the average SWE of the original sampling frequency for each day of sampling (2019-10-31 to 2020-04-26).

2020b) (in the following referred to as P2PS) (Tab. 1). The PWD22PS was installed at 22 m and the P2PS at 24 m above the water line. On the ice, in "Met City" (Fig. 2, in the following, referred to as MC), three precipitation sensors were installed: 1) an OTT Parsivel[2] (P2MC), installed at 1.5 m nominal height above the snow surface, surrounded by a double-alter shield, 2) a PWD22 (PWD22MC), installed at 2 m nominal height, unshielded and 3) an OTT Pluvio[2] L (Wang et al., 2019b; Bartholomew, 2020b), shielded by a double-alter shield and installed at 1 m above the snow (in the following referred to as Pluvio2). Different ARM data levels of the devices are given, where a1 means "calibration factors applied and converted to geophysical units" and b1: "QC checks applied to measurements".

Optical devices evaluated here are the Vaisala PWD22 and the OTT Parsivel[2]. However, the measurement technique and the process of estimating snowfall rates are different. The Parsivel[2] is a laser-disdrometer that processes the voltage signal changes due to light extinction when a hydrometeor falls through the laser-beam. It has an effective measuring area of $54\,\text{cm}^2$ to esti-

mate hydrometeor size and velocity (Löffler-Mang and Joss, 2000). The hydrometeors are classified into size classes which can be used to investigate the particle size distribution. The precipitation type is determined by device-internal spectral signature comparison, where the spectra are determined empirically. Based on particle size, velocity, and estimated precipitation type, device-internal software computes a snowfall estimate. No details are known about the exact formula used by the manufacturer

for the snowfall estimate. Its accuracy is given by the manufacturer as $\pm\,20\,\%$ with an intensity range of 0.001 to $1200\,\mathrm{mm\,h^{-1}}$ (Tab. 1). The calibration was conducted in the manufacturer's laboratory and therefore no calibration was needed in the field.

The PWD22 consists of several sensors that are used to compute the snowfall rate: The two core sensors are a transmitter-receiver combination, where the transmitter emits pulses of near-infrared (NIR) light. The receiver on the other side measures

the scattered part at $45°$ of the light beam from the emitted signal (sampling volume $100\,\mathrm{cm^3}$). Rapid changes in the scatter signal between transmitter and receiver are used to compute precipitation intensity. The sampling volume allows for the detection of single crystals and aggregates of snow crystals (snowflakes). Furthermore, the PWD22 is equipped with a heated RAINCAP rain sensor, which produces a signal proportional to the amount of water on the sensing element. By means of the ratio from sample volume and water content determined with the RAINCAP sensor, precipitation types are distinguished. In the tube

between the transmitter and receiver, another temperature sensor (thermistor) is installed. The detected temperature is used to select the default precipitation type. When frozen precipitation is detected, the PWD22 software multiplies optical intensity with a scaling factor, determined from RAINCAP and optical intensities from the receiver to estimate snowfall intensity as SWE per time unit (Vaisala, 2004). The manufacturer does not provide a value for accuracy, however the intensity measuring range is given as 0.00 to $999\,\mathrm{mm\,h^{-1}}$. There is no calibration principle known for the field but the manufacturer mentions

comparisons with closeby reference gauges as calibration method.

The only device that we compare here that uses a weighing principle is the OTT Pluvio[2]. The instrument's core is a sealed load cell that continuously measures the weight of the precipitation falling into the entry of the bucket. The installed variant was an OTT Pluvio[2] L Version 400, with a collecting area of $400\,\mathrm{cm^2}$ and a recording capacity of 750 mm of precipitation. Its

accuracy is given by the manufacturer as $\pm\,0.1\,\mathrm{mm\,min^{-1}}$ or $\pm\,6\,\mathrm{mm\,h^{-1}}$, or $\pm\,1\,\%$ and its intensity range as $\pm\,6\,\mathrm{mm\,h^{-1}}$ or 0.1 to $30\,\mathrm{mm\,h^{-1}}$. No calibration for the OTT Pluvio[2] is needed in the field as it was delivered calibrated by the manufacturer. However, calibration weights were used to test for accuracy.

The data streams were downloaded from the ARM data archive (https://adc.arm.gov/) and scanned for quality control flags. Values with timestamps that correspond to flags indicating maintenance time, suspicious or incorrect values were discarded.

**2.4.2    Snowfall retrievals from the Ka-band ARM Zenith Radar.**

Snowfall was retrieved from the $\mathrm{K}_a$-Band ARM Zenith Radar (KAZR) (Widener et al., 2012; Lindenmaier et al., 2019) that was installed on a container at the bow of RV Polarstern. Using a radar snowfall retrieval allows investigating snowfall continuously, and eliminates impacts on gauges such as acceleration effects of wind that result in undercatch or overestimation due to blowing snow particles. The KAZR operated at approximately 35 GHz. We computed the snowfall rate S $[mm\,h^{-1}]$ according to the

**Table 1.** Summary of validated installed precipitation sensors and radar during MOSAiC as well details about the ERA5 reanalysis (PS: RV Polarstern, MC: Met City, DA: double alter shield, WL: Water line, DFIR: Double Fence Intercomparison Reference).

| Device /Reanalyis | Loc | Abbreviation | Nominal height Snow/WL (m) | Shield | Data reference | ARM data level | Accuracy | Intensity range | Selected existing snowfall validations / Reference setup | Calibration |
|---|---|---|---|---|---|---|---|---|---|---|
| Vaisala PWD22 | PS | PWD22PS | 22 (WL) | - | Kyrouac and Holdridge (2019) | b1 | - | 0.00 - 999 mm/h | -32 % from median / DFIR (Wong, 2012a) +33 % from median / manual (Boudala et al., 2016) | Reference comparison |
| OTT Parsivel[2] | PS | P2PS | 24 (WL) | - | Shi (2019) | b1 | ± 20 % | 0.001 - 1200 mm/h | +46/+54 % from median / DFIR (Wong, 2012a) +24/+29 % from median / DFIR (Wong et al., 2012b) | manufacturer |
| KAZR | PS | KAZR | 14 (WL) | - | Lindenmaier et al. (2019) | a1 | - | - | 23 % relative bias / shielded Nipher gauge (Matrosov et al., 2008) | - |
| Vaisala PWD22 | MC | PWD22MC | 2 (Snow) | - | Kyrouac and Holdridge (2019) | b1 | - | 0.00 - 999 mm/h | see above | Reference comparison |
| OTT Parsivel[2] | MC | P2MC | 1.5 (Snow) | DA | Shi (2019) | b1 | ± 20 % | 0.001 - 1200 mm/h | see above | manufacturer |
| OTT Pluvio[2] | MC | Pluvio2 | 1 (Snow) | DA | Wang et al. (2019b) | a1 | ± 6 mm/h ± 0.1 mm/min ± 1 % | 6 - 1800 mm/h | -63 % from median (max) / DFIR -14 % from median (min) / DFIR (Wong, 2012a) | manufacturer / calib. weights |
| ERA5 | - | - | - | - | Hersbach et al. (2020) | - | - | - | + 62.8 mm cum. SWE / snow buoys (Wang et al., 2019a) slight underestimate in summer / CloudSat (Cabaj et al., 2020) overestimate in other months | - |

power law

$$Z_e = a\,S^b \tag{7}$$

where $Z_e\,[mm^6\,m^{-3}]$ is the radar equivalent reflectivity factor and $a$ and $b$ are empirical coefficents. We chose $a = 56$ and $b = 1.2$ as these were found to be good average values for dry snowfall at this radar frequency, and no significant riming was observed (Matrosov, 2007; Matrosov et al., 2008).

Near-field radar measurements can suffer from a variety of issues, such that snowfall retrievals typically must be applied to radar signals that are elevated above the surface. To find an appropriate KAZR range gate to extract snowfall rates, we plotted the cumulative sums of SWE based on KAZR-derived snowfall from reflectivity measurements at different range gates (Fig. 7). The first range gate of 100 m did not yield any measurements, while at 130 m reflectivities were too low. From Fig. 7 we see that the differences in the cumulative snowfall from range gates between 220 m, and 280 m are the least. The decrease

of computed snowfall with height beyond 280 m is probably due to very low cloud heights in winter (Jun et al., 2016), such that snowfall would get underestimated as these range gates are often at higher elevations within the clouds or even beyond the cloud top. We found largest snowfall rates for the 280 m range gate, thus we chose it as range gate from which we extracted the snowfall retrievals. However, based on this simple analysis, the potential differences in snowfall based on this choice of range gate are on the order of about 10 %. With an instrument elevation of 14 m asl, the elevation of the extracted snowfall rates is

294 m asl.

## 2.5 Atmospheric flux station data

A meteorological tower of 10 m height was installed on the ice 558 m away from RV Polarstern at about 60° off the bow of the vessel in the middle of October 2019. However, due to ice dynamics, by the end of leg 3 (beginning of May 2020), the distance was only about 334 m while the direction from the ship stayed approximately the same (Fig. 2). At nominal levels $z = 2\,\text{m}$, 6 m

and 10 m above the snow, three-dimensional wind $(u, v, w)$ and temperature were measured at high frequency with METEK uSonic-3 Cage MP anemometers (METEK, 2019), while on the same elevation levels, relative humidity and temperature were

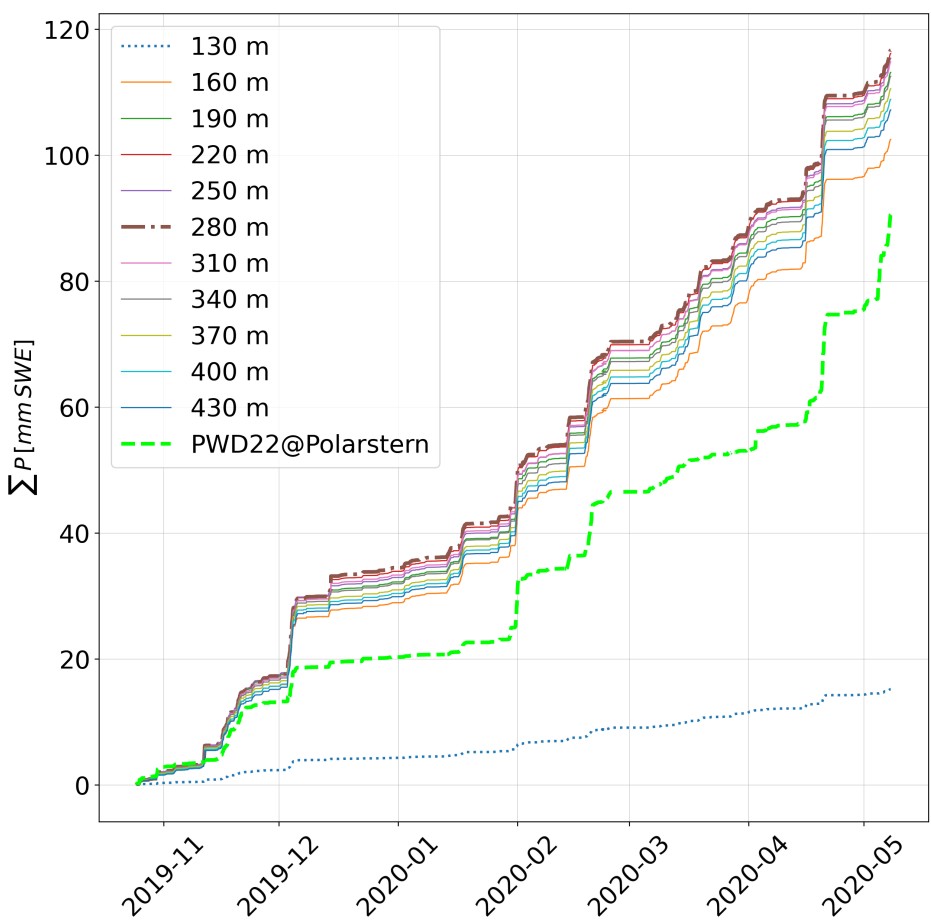

**Figure 7.** Computed cumulative snowfall for different KAZR range gates and PWD22PS.

measured with Vaisala HUMICAP Humidity and Temperature HMT330 sensors (Vaisala, 2009). The University of Colorado / NOAA surface flux team carried out the post-processing and computed turbulent fluxes, such as momentum flux and turbulent heat fluxes, mixing ratio, or friction velocity. Wind vectors were corrected, i.e., processed wind directions are according to geographic true north. We used wind velocity, wind direction, computed latent heat flux, friction velocity, relative humidity, the temperature at 2 m, and temperature of the snow surface from the described dataset (Cox et al., 2021).

### 2.6 Drifting and blowing snow mass flux

On the meteorological tower described under Sec. 2.5, two snow particle counters (SPCs) (Sato et al., 1993) were installed. The devices continously detect number and sizes of snow particles which are transported through a laser beam. The devices rotate with very low friction on a vertical axis and mounted wind vanes at the back of the sensor keep the laser beam 90° towards the wind. One SPC was installed at about 0.1 m (SPC1104) and one at 10 m (SPC1206) above the snow. The lower

SPC1104 ran with only a few interruptions from 2 December 2019 until 7 May 2020 (data availability for this period 96.6 %). The upper SPC1206 ran with only a few interruptions from 14 October 2019 until 7 May 2020 (data availability for this period 94.9 %). One bigger data gap for the SPC1206 was between 17 November 2019 03:05:00 UTC and 18 November 11:15:00 UTC because there was a power interruption due to sea ice dynamics resulting in broken power lines. Note that the SPC1206 data is still under quality control and the correctness of the mass flux values should be questioned at this time. Nevertheless, the comparison with the lower SPC1104 gives us an order of magnitude of the mass flux to be expected at the height of 10 m compared to lower heights. Therefore, we have made the comparison here and would like to point out that the values of the SPC1104 at 0.1 m are more meaningful at this point in time.

To determine periods, where snow transport and erosion has occurred, horizontal mass flux $\left(kg\,m^{-2}\,s^{-1}\right)$ for both SPCs were computed as (Sugiura et al., 2009):

$$Q_{SPC} = \frac{\pi\,\rho_p}{6} \sum_{n=1}^{64} S_n\,N_n\,D_n^3 \tag{8}$$

where $\rho_p$ is the density of a drifting snow particle, which we assumed here as the density of ice $\rho_p = 917\,kg\,m^{-3}$, $S_n$ is the shape factor of snow particles of the $n$-th class, which we assumed as 1 here, $N_n$ is the particle flux of the $n$-th class $\left(m^{-2}\,s^{-1}\right)$, which is the number of particles per class passing the SPC sensor area $A_s$ in a second, and $D_n$ is the diameter of a drifting snow particle of the $n$-th class $(m)$.

It is likely that the distance between sensor and snow cover varied over time since installation on 2 December 2019 due to deposition of new snow. In any case, given these uncertainties, to determine potential drifting snow periods for periods where the SPC might fail, the critical friction velocity for snow particles was calculated as (Bagnold, 1941):

$$u_{*t} = A \cdot \sqrt{\frac{\rho_{ice} - \rho_{air}}{\rho_{air}} g\overline{d}} \tag{9}$$

where $A$ is a threshold parameter and is here assumed as 0.18 as found by Clifton et al. (2006) for drifting snow initiation, $\rho_{ice} = 917\,\mathrm{kg\,m^{-3}}$ is the density of ice, $\rho_{air}$ is the density of air, $g = 9.81\mathrm{m\,s^{-1}}$ is gravity acceleration on earth and $\overline{d}$ is an average particle diameter, which we assumed as $260\,\mu m$ which was found as lowest particle diameter on the surface where snow transport was observed by Clifton et al. (2006). $\rho_{air}$ could be retrieved from the meteorological tower data. The computed thresholds were applied to computed $u_*$ from the tower. If $u_* > u_{*t}$, particles begin to get lifted from the ground and drifting snow flux is initiated.

The mass decrease computed with the HS-SWE function from the transect is temporally compared against computed cumulative snow mass flux from the snow particle counters. Note that the cumulative horizontal mass flux is only an indicator for the strength of the erosion, but cannot be translated into actual eroded mass. To distinguish in the text between computed SWE decrease of the snow cover and cumulative mass flux and to avoid confusion, we keep the designation SWE for the snow cover but use $\mathrm{kg\,m^{-2}}$ for cumulative mass flux in the following, although SWE has the same units.

## 2.7 ERA5 mean snowfall rates

For the drift track coordinates of RV Polarstern, shown in Fig. 1, we extracted ERA5 (Hersbach et al., 2020) mean snowfall rates, which is the sum of the convective and large-scale snowfall in ERA5. While the large-scale snowfall is generated from the cloud scheme in the ECMWF Integrated Forecasting System (IFS) (ifs, 2020), the convective snowfall is generated from the IFS convection scheme. The resolution for ERA5 over the sea is 0.28125° x 0.28125°, which is about 31 x 31 km. Hence, the extracted snowfall rate from ERA5 for the drift track does not refer to points but represents an averaged value over these grid cells closest to the drift track coordinates. The purpose here is to compare the ERA5 mean snowfall against snow cover SWE and sensors in this study.

## 2.8 Sensor and reanalysis comparison method

We computed the average SWE for the northern- and southern loop (Tab. 2) as we expect this combination of deformed SYI, remnant SYI, and FYI is more representative for an overall snow accumulation estimate than choosing a snow deposition for one of these ice types. Additionally, in section 3 and Fig. 9 we will show, that change of average SWE along a several hundreds of meters long section of the whole transect is over 200% from the average SWE along the whole northern transect, compared before and after a drifting snow event. This confirms the need for as long transect sections as possible to find an average snow mass value for an area that can be representative for the MOSAiC ice floe. Indeed it is impossible to determine at this point with this data set we made use of whether only the SWE derived from the northern- or southern- or the average of loops, is the best choice to evaluate snowfall. A snow height difference dataset based on laser scanners of an area that includes both, northern- and southern transects and an area beyond that, could be used to validate transect snow depth. However, we do not make use of such a dataset here. We will demonstrate in the coming sections that initial average SWE on the northern loop is about twice the value of the average SWE on the southern loop.

The SWE increase until January 2020 is much faster on the southern loop, hence, we see a different accumulation rate depending on whether we measure snow depth on SYI (northern loop) or FYI (southern loop). Indeed, what "most representative" means, also strongly depends on the horizontal extent of snowfall and wind patterns, i.e. the total accumulated snow mass that has fallen over a certain area but is re-deposited due to wind. This is a problem we are not able to consider in this study but can potentially be solved by computing snow mass based on the difference of airborne- or terrestrial laser scan derived heights. The reason why we decided to use an SWE average of Northern and Southern loop as reference, is that the MOSAiC ice floe consisted to large parts of these two ice types.

This average computed snow cover SWE serves then as our reference for the precipitation sensors and ERA5 snowfall. Note that for the averaging process, data were discarded when only the northern- or southern transect loop was measured on one day, except when the temporal distance between the measurement of northern and southern loop was short and there was no snowfall in between. This is only the case for 24 (northern) and 26 (southern) April. Hence, the transect averaged SWE starts on 31 October 2019 and ends on 24/26 April 2020, with a significant reduction of days of sampling, compared to all days of transect sampling available (Tab. 2). RMSE between snow cover SWE and sensor- and reanalysis estimated SWE was com-

puted for the time period where no SWE decrease in between the days of the transect sampling has been detected by means of the computed snow cover SWE, which is all days before and including 20 February 2020. Here we used n = 10 days for subtracting SWE, which results in n = 9 days for error comparison. The RMSE is computed as mm and always refers to the precipitation sum between days of transect sampling.

To discuss the erosion influence on potential discrepancies of snow cover SWE and sensor estimated snowfall in more detail, in addition, RMSE was computed for days after time periods where no significant amounts of horizontal mass flux were detected with the SPCs, i.e. where no erosion between two days of transect sampling was expected. These detected drifting snow periods until 20 February were 3 - 5 December, 19 December 2019, 30 Jan - 2 February and 18-20 February 2020. Hence, in this case, 5 December 2019, 6 February and 20 February 2020 were discarded from transect SWE time series for evaluation, hence n = 6 days, which are the days before and after the drifting snow events, were left over for comparison with sensors and ERA5. This results in n = 5 pairs for error calculation. The drifting snow free periods are marked as yellow areas in the graph of Fig. 11f. Note that due to the strong cumulative aspect (i.e. we compare snowfall that is accumulated always between the days of transect measurements), the difference is naturally reduced when reducing the sample number. The reduction of validation pairs reduce the significance of the comparison, however, we additionally inter-compare the error change of the sensors/ERA5 which may strengthen the significance considering the role of wind.

## 3 Results

### 3.1 Snow mass accumulation and decrease

Fig. 8a shows the derived SWE evolution as box- and whisker plots for the northern and the southern transect. The initial average SWE values for the northern loop (66 mm on 31 October) are naturally higher than for the southern loop (32 mm on 31 October), as the northern transect was situated mostly on deformed SYI. The value for the northern loop decreased to 65 mm until 5 December, while average SWE for the southern loop increased, by 29 mm to a similar value of 66 mm between 14 November and 5 December 2019. The SWE on both transect loops increased to 92 mm on the northern loop and 80 mm on the southern loop, between 5 December 2019 and 20 February 2020. From then on, there was a decrease observed in both loops, with a minimum of 79 mm on the northern loop on the 6 April 2020 and a minimum of 73 mm on the southern loop on 5 March 2020. On both transects, SWE increased afterward, to 90 mm on the northern loop by 24 April 2020 and 81 mm on the southern loop by 26 April 2020. Hence, even though the initial SWE over the remnant SYI and FYI (southern loop) was approximately only half of the value on the northern loop, it reached 90 % of the snow mass of the northern loop by the end of the accumulation period.

In the following, we present detailed results about snow mass decrease. Notably, we find a net mass decrease, computed with the HS-SWE function, of 9.5 mm for the northern transect loop between 27 February and 20 March. The mass increased again to 90 mm between 20 March and 24 April 2020. The maximum on the northern loop is reached with 94 mm on 7 May 2020. This however, is not comparable with the southern loop as the southern transect time series only last until end of April 2020. The SWE maximum on the southern loop is found as 81 mm on 26 April 2020. The time from 20 February to 20 March

**Table 2.** Used transect days of sampling, and specifications: Transect N/S refers to Northern/Southern Transect, averaged for validation refers to the days where the transect SWE was averaged as shown in Fig. 8, RMSE with drifting snow describes all days where SWE was subtracted from the from the respective day before, while all days where subtracted in a row, RMSE without drifting snow / pair refers to the days used for validation where no or low drifting snow mass flux was detected in between while the respective days after which subtraction was performed consecutively are listed as well. The corresponding RMSE values including regression lines are shown in Fig. 12. The comparison with bulk SWE and SMP refers to the dates where direct comparisons are made with HS-SWE function derived SWE along the transect (Fig. 5.

| Date | Transect N / S | Averaged for validation | RMSE with drifting snow | RMSE without drifting snow / pair | Comparison with bulk SWE and SMP |
|---|---|---|---|---|---|
| 2019-10-24 | N | - | - | - | - |
| 2019-10-31 | N / S | yes | yes (start) | - | - |
| 2019-11-07 | N / S | yes | yes | yes (start) / 2019-11-14 | - |
| 2019-11-14 | N / S | yes | yes | yes / 2019-11-07 | yes |
| 2019-11-21 | N | - | - | - | yes |
| 2019-11-28 | N | - | - | - | yes |
| 2019-12-05 | N / S | yes | yes | - | - |
| 2019-12-19 | N | - | - | | - |
| 2020-01-02 | N / S | yes | yes | yes / 2019-12-05 / 2020-01-09 | - |
| 2020-01-09 | N / S | yes | yes | yes / 2020-01-02 / 2020-01-16 | - |
| 2020-01-16 | N / S | yes | yes | yes / 2020-01-09 / 2020-01-30 | - |
| 2020-01-30 | N / S | yes | yes | yes (end) / 2020-01-16 | - |
| 2020-02-06 | N / S | yes | yes | - | - |
| 2020-02-20 | N / S | yes | yes (end) | - | - |
| 2020-02-27 | N / S | yes | - | - | - |
| 2020-03-05 | N / S | yes | - | - | - |
| 2020-03-20 | N | - | - | - | - |
| 2020-03-26 | N | - | - | - | - |
| 2020-03-30 | S | - | - | - | - |
| 2020-04-06 | N | - | - | - | - |
| 2020-04-16 | N | - | - | - | - |
| 2020-04-24 | N | yes (avg with 2020-04-26) | - | - | - |
| 2020-04-26 | S | yes (avg with 2020-04-24) | - | - | - |
| 2020-04-30 | N | - | - | - | - |
| 2020-05-07 | N | - | - | - | - |

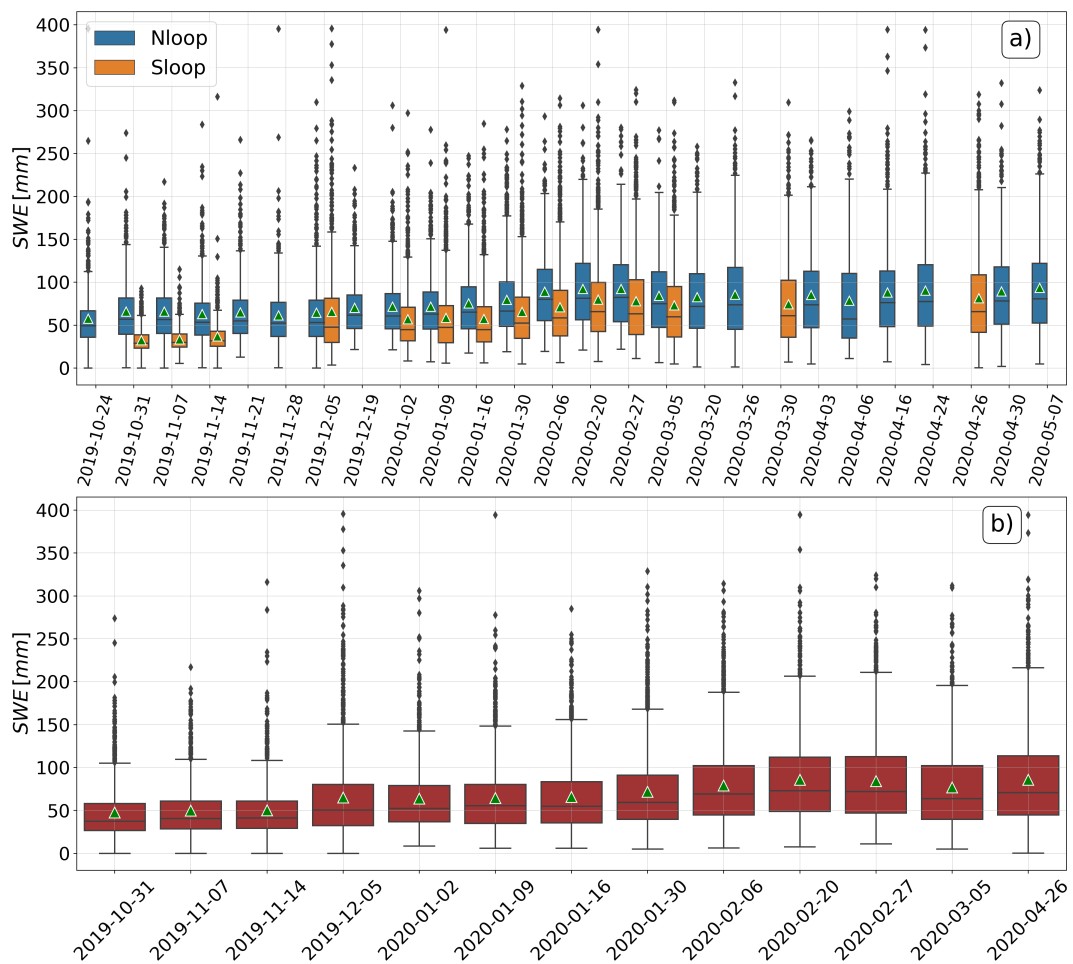

**Figure 8.** a) Box- and whisker plots for SWE estimates along the northern ("Nloop") and southern ("Sloop") transect, respectively. b) Shows box- and whisker plots for averages of northern- and southern transect loop. Horizontal lines show the median, green triangles the average, the boxes show the interquartile ranges (IQR) (25 - 75 %), the whiskers represent 1.5 times the upper- and lower values of the IQR. The dots represent outliers that are beyond 1.5 times the IQR. Note the different dates between a) and b) as data where only data for one loop was available was discarded for computation of b). The exception is the 24, and 26 April 2020 as the temporal distance was so close that these were averaged, too.

exactly falls into the period, where 1) the discrepancy between cumulative SWE from the precipitation sensors and SWE from the transect becomes large (Fig. 10b, Fig. 11a) and 2) where about 45 % ($1.977 \cdot 10^6$ kg m$^{-2}$) of the total cumulative horizontal snow mass flux at 0.1 m above the surface over the whole measuring period of the SPC1104 has occurred (Fig. 11f). That means, over 45 % of drifted snow-mass appeared on only 19 % (30/158) of the days from the whole measuring time of the lower SPC. The period is marked as green shaded areas in Fig. 11. Most distinct in this period was the event on 24 - 25 February (marked as red shaded areas in Fig. 11), during which $1.014 \cdot 10^6$ kg m$^{-2}$ of cumulative mass flux was detected

with the lower SPC - which is 23 % of the total detected cumulative mass flux on 1.3 % of the days the device was running.

During this storm, a maximum peak of around $11\,\mathrm{m\,s^{-1}}$ was detected in the 1-h averaged wind speed data at 2 m above the ice, which means that the measured peak at shorter time intervals must have been higher. We computed the sum of the detected mass decrease that occurred between days of sampling and that is driven by erosion (but does not reflect the total eroded mass), for the northern loop as $\Delta SWE = -24\,\mathrm{mm}$ (between 31 October 2019 - 26 April 2020) and for the southern loop as $\Delta SWE = -16\,\mathrm{mm}$ (between 31 October 2019 - 24 April 2020). Fig. 9 shows the SWE for the same section of the northern

loop transect as in Fig. 5 (268 m length), for 20 February and 5 March 2020, which are before and after a drifting snow event. As the section on 5 March had twice the average horizontal Magnaprobe sampling distance compared to 20 February, and for better illustration, a simple moving average with a window of $n = 4$ was applied to the section on 20 February while a moving average with a window $n = 2$ was applied to the data from 5 March. A significant re-distribution due to wind is recognizable. During the same period, the average decrease for this section was 12 mm SWE (from 82.6 mm to 70.1 mm - a relative value of

15 %), while for the whole northern loop the average decrease was 8 mm (from 87.3 mm to 79.5 mm - a relative value of 9 %).

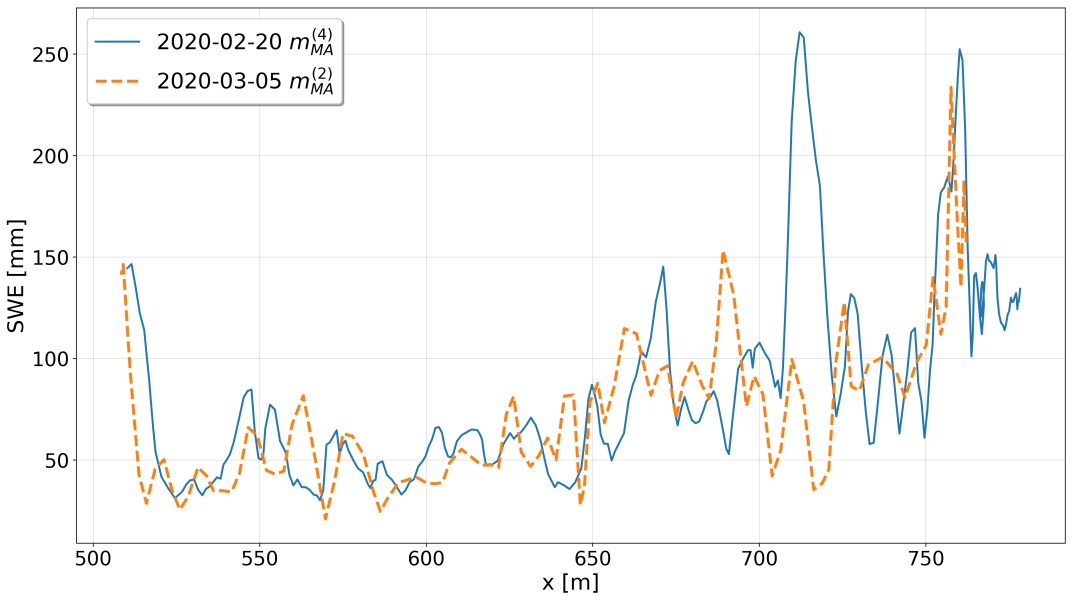

**Figure 9.** Retrieved SWE from same Magnaprobe section as shown in Fig. 5 on 20 Februrary before a strong drifting snow event that occurred on 24 - 25 February and mass distribution on 5 March 2020 after the drifting snow event. The superscript numbers in brackets in the legend correspond to the count of numbers of the moving median window used for plotting.

Finally, we present results of the average SWE of the northern- and southern loop (Fig. 8b). The average initial SWE value on 31 October 2019 was 48 mm and increased to 86 mm on 24/26 April 2020. Hence, we estimate the total mass increase over time as about 38 mm. Over the transect average, the SWE decrease during the snow transport event was 5.5 mm on 24 - 25 February 2020.

## 3.2 Precipitation sensor and radar snowfall retrieval comparisons


To compare with the different estimates of snowfall, the cumulative SWE values were examined for each approach. The plot of cumulative snowfall between 31 October 2019 and 7 May 2020 without any corrections applied can be seen in Fig. 10a. The snowfall rates deviate heavily from one another between precipitation data source and locations. The P2PS shows the highest cumulative snowfall, while the P2MC shows the lowest, although with limited data availability (Bartholomew, 2020b). As a

result of the limited availability and wide spread between these two identical systems operated at different locations, we no longer consider the P2MC. The PWD22PS shows the lowest cumulative snowfall, with 97.6 mm while the highest is estimated by the P2PS (290.3 mm). It also stands out that the cumulative sum of ERA5 (110 mm) by 7 May 2020 is very similar to that of the snowfall estimated from the KAZR (114 mm).

Fig. 10b compares the northern, southern, and average transect loop SWE values with the uncorrected cumulative snowfall from a subset of the sensors and the ERA5 mean snowfall. The PWD22PS is well in line with SWE from the snow cover until mid-end of February 2020. Afterwards, more snow over the ice was eroded, which is indicated by high horizontal drifting and blowing snow mass flux measured with both SPCs (Fig. 11f). After mid-February, the snow cover SWE did not increase significantly and instead stagnates, while the sensors indicate periodic snowfall. Thus, the discrepancy between sensor snowfall rates

and snowpack SWE became larger. RMSE with respect to SWE difference is shown for snowfall sensors, KAZR and ERA5, (Fig. 12). We consider the case first where snowdrift time periods are included in the evaluation (i.e. all days of sampling until and including 20 February 2020). For this period, a SWE increase of the snow cover of about 37 mm was detected (Fig. 12a, cumulated within the intervals in between n = 9 days). Fig. 12b, in contrast, shows RMSE computed only for days when no drifting or low drifting snow occurred (n = 6 days). For this time period, an increase of 13.7 mm SWE was detected for the

transect. Considering the first case (n = 9 days), all sensors appear to overestimate. However, as expected, this indicates, that erosion occurred in the time periods between the days the transects were sampled, which leads to a systematic positive bias of the sensors. For this case the PWD22PS is most similar to the SWE (RMSE = 2.06 mm), followed by ERA5 (3.37 mm), the KAZR (4.7 mm), Pluvio2 (6.78 mm), PWD22MC (8.74 mm) and P2PS (26.39 mm).

However, in the second case (n = 6 days), the differences are reduced significantly for all devices and ERA5, too. In this case,

Pluvio2 shows the closest comparison to the SWE (RMSE = 1.72 mm). PWD22PS shows good agreement (RMSE = 1.96 mm) with a tendency towards underestimation. It reveals that also ERA5 performs well (RMSE = 2.88 mm), but with an overestimation tendency, as all validation pairs were positively biased with an average of 2.0 mm. Besides ERA5, the KAZR (RMSE = 3.55 mm), systematically postively biased with 2.4 mm, show least RMSE decrease comparing to the case using n = 9 days, which is very likely the result of the fact that both ERA5 and KAZR are not wind-vulnerable in contrast to the other sensors.

The largest difference of the sensors relative to the snow cover is found for P2PS (RMSE = 3.2 mm).

Note that due to the strong cumulative aspect (i.e. we compare snowfall that is accumulated always between the days of transect measurements), the difference is naturally reduced when reducing the sample number. Nonetheless, the fact that there is an overall tendency towards a decrease in the apparent overestimation of the sensors relative to the SWE, indicates that erosion

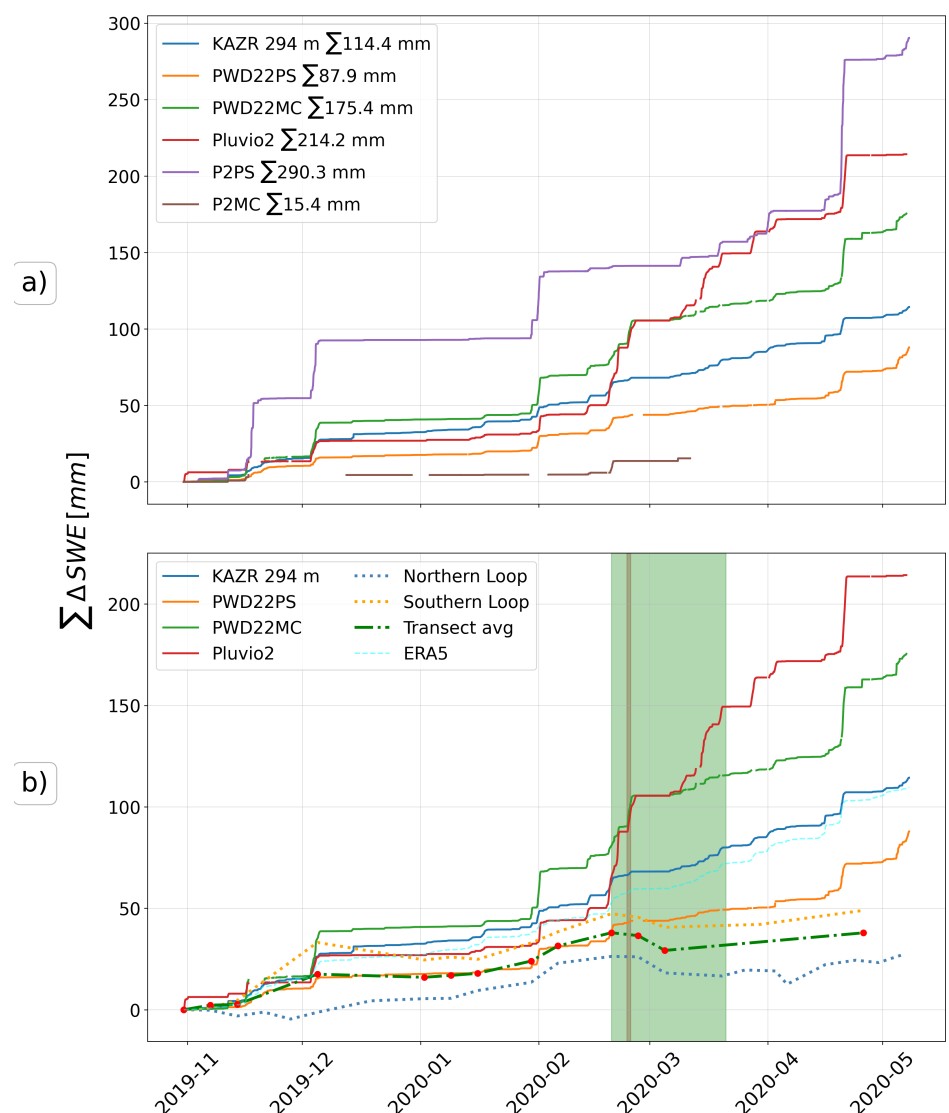

**Figure 10.** a) Cumulative snowfall for different installed precipitation sensors during MOSAiC from 31 October 2019 to 7 May 2020. b) Sensors, ERA5 estimates and SWE of the snow cover. The red dots show the days on which both transect loops were sampled. The red shading shows the time period of the strong drifting snow event on 24 - 25 February 2020. The green shaded area mark the strong drifting snow period where 45 % of all cumulative horizontal mass flux was detected (Fig. 11).

likely did occur before the days eliminated in Fig. 12b. For the PWD22PS, the RMSE is only reduced by about 2.5 %, for ERA5 by 13 % and the KAZR by 23.7 % while for P2PS the RMSE was reduced by 88 % and for PWD22MC reduced by 69 %. For the Pluvio2, the difference was reduced to 26 % of its initial value. While the apparent overestimation of the sensors in Fig. 12a is likely due to erosion (and hence strongly biased), these different magnitudes of RMSE reduction (Fig. 12b) suggest that

PWD22PS is less affected by overestimation due to high wind speeds that accompany blowing snow, compared to PWD22MC or Pluvio2, both of which were installed near the surface, but also compared against P2PS, which is a device known for wind vulnerability towards overestimation of snowfall.

In summary, if we only consider time periods without drifting snow, Pluvio2 and PWD22PS compare most favorably with

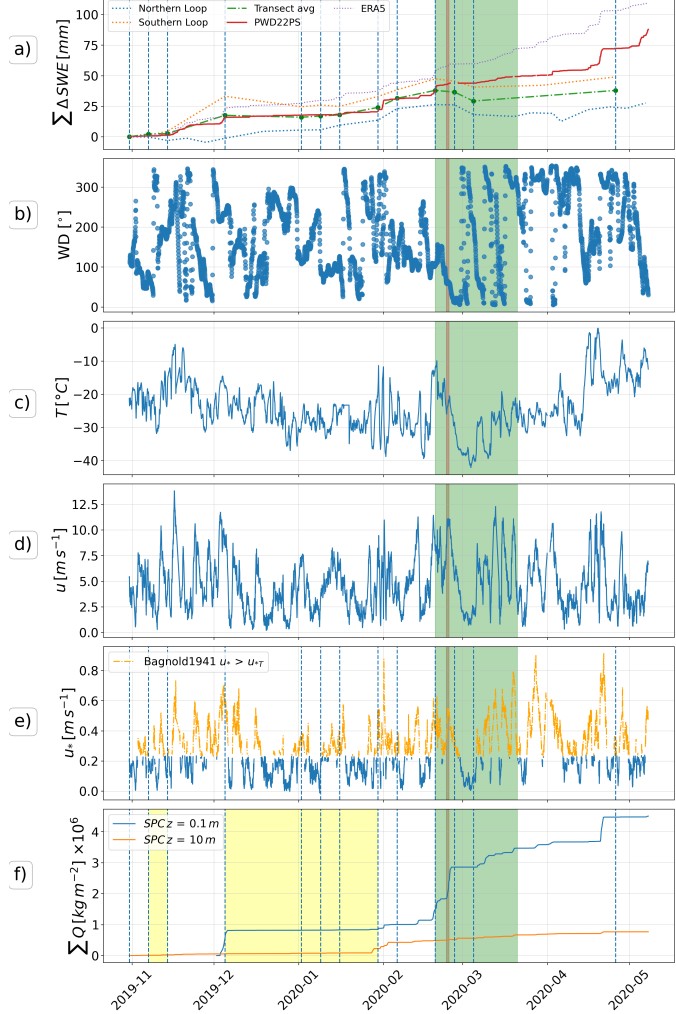

**Figure 11.** Time series from 31 October 2019 to 7 May 2020 for a) estimated SWE from the transect and HS-SWE model as well as cumulative snowfall from ERA5, cumulative snowfall from PWD22 on Polarstern and KAZR derived snowfall rates. b) wind direction at 2 m, c) air temperature at 2 m above the snow d) wind speed at 2 m height, e) computed friction velocity threshold for snow transport after (Bagnold, 1941), f) cumulative horizontal mass flux with the snow particle counter at 0.1 m above the snow and at 10 m height, respectively. The green shaded areas mark the strong drifting snow period where 45 % of all cumulative horizontal mass flux was detected. The vertical blue dashed lines mark the days where both transect loops were sampled. The yellow shaded areas mark the time periods no to very low mass was detected by means of the snow particle counters.

SWE, however with reasonable results for ERA5, KAZR, PWD22MC and P2PS, as well. When also considering high wind speeds and blowing snow, the PWD22PS still appears to compare most favorably with SWE, while especially P2PS, PWD22MC and Pluvio2 appear to be most negatively affected by high wind speeds.

Taken together, we detected five significant snowfall events. If we use PWD22PS as reference, we find for 3 - 5 December 2019: $\approx 5.5$ mm, 30 January - 3 February 2020: $\approx 10$ mm, 18 - 21 February 2020: $\approx 8.5$ mm, 16 - 21 April 2020: $\approx 16.5$ mm, 4 - 7 May 2020: $\approx 14$ mm. Hence, about 54 mm of snow fell during events, while the other 33 mm fell in between, e.g. as trace precipitation or diamond dust.

To better illustrate the blowing snow influence on sensors that is already suggested by Fig. 12, we made scatter plots of snowfall rates from different sensors with respect to the PWD22PS. Fig. 13 shows a scatter plot for the short time of two days between 24 and 25 February, where high drifting snow mass fluxes were detected. We can clearly see that the Pluvio2 (Fig. 13b) and the PWD22MC (Fig. 13d) strongly overestimate snowfall relative to PWD22PS, while P2PS (Fig. 13a) and KAZR (Fig. 13d) stay largely unaffected and only measure trace precipitation of 0.1 to 0.2 mm h$^{-1}$. This becomes even clearer when we look at snowfall rates of Pluvio2 (Fig. 14a) and PWD22MC (Fig. 14b) versus the horizontal mass flux detected by the SPCs.

Scatter plots for the whole period (31 October 2019 - 7 May 2020) for different sensors versus horizontal mass flux reveal the influence of drifting and blowing snow, too (Fig. 15). Pearson correlation coefficients show medium positive correlations for mass flux and snowfall from Pluvio2 at SPC height at 0.1 m, (Fig. 15b, r = 0.55 - 0.58) and PWD22MC (Fig. 15d, r = 0.54 - 0.55) while a weak negative correlation is observed for P2PS (Fig. 15a, r = -0.19) and for PWD22PS (Fig. 15c, r = 0.26). These results indicate that instruments collocated on the ice at about 1 - 1.5 m height are much more affected by drifting and blowing snow than instruments installed on Polarstern at 22 m height. We compare ERA5 mean snowfall rates against the snow cover SWE and against PWD22PS. As described above, when we consider the comparisons illustrated in Fig. 12b, ERA5 shows reasonable results with a relatively low RMSE of 2.8 mm and an overestimation tendency. Assuming PWD22PS as reference, ERA5 shows an overall good timing of the snowfall events (Fig. 10b, Fig. 11a). As for the transect SWE validation, it overestimates snowfall relative to PWD22PS, too, in this case systematically and with an acceleration of the positive bias from the end of February on. This leads to an overestimation (relative to PWD22PS) of the total accumulation of almost 22 mm (+25 %) by the end of the investigation period. We computed the RMSE for the snowfall rate relative to PWD22PS as 0.06 mm h$^{-1}$ for the whole time period from 31 October to 7 May 2020.

## 4 Discussion

### 4.1 Snow mass balance

With the fitted HS - SWE function, we were able to retrieve the SWE of snow cover over the ice for the transect loops. We could show that comparing the average SWE of a northern loop section with 268 m length (Fig. 9) with the average SWE for the whole northern loop, the SWE change due to a drifting snow event was different by more than 100 %. This shows the need for sampling with large spatial extents which was one reason - besides including both characteristic ice types for the ice floe

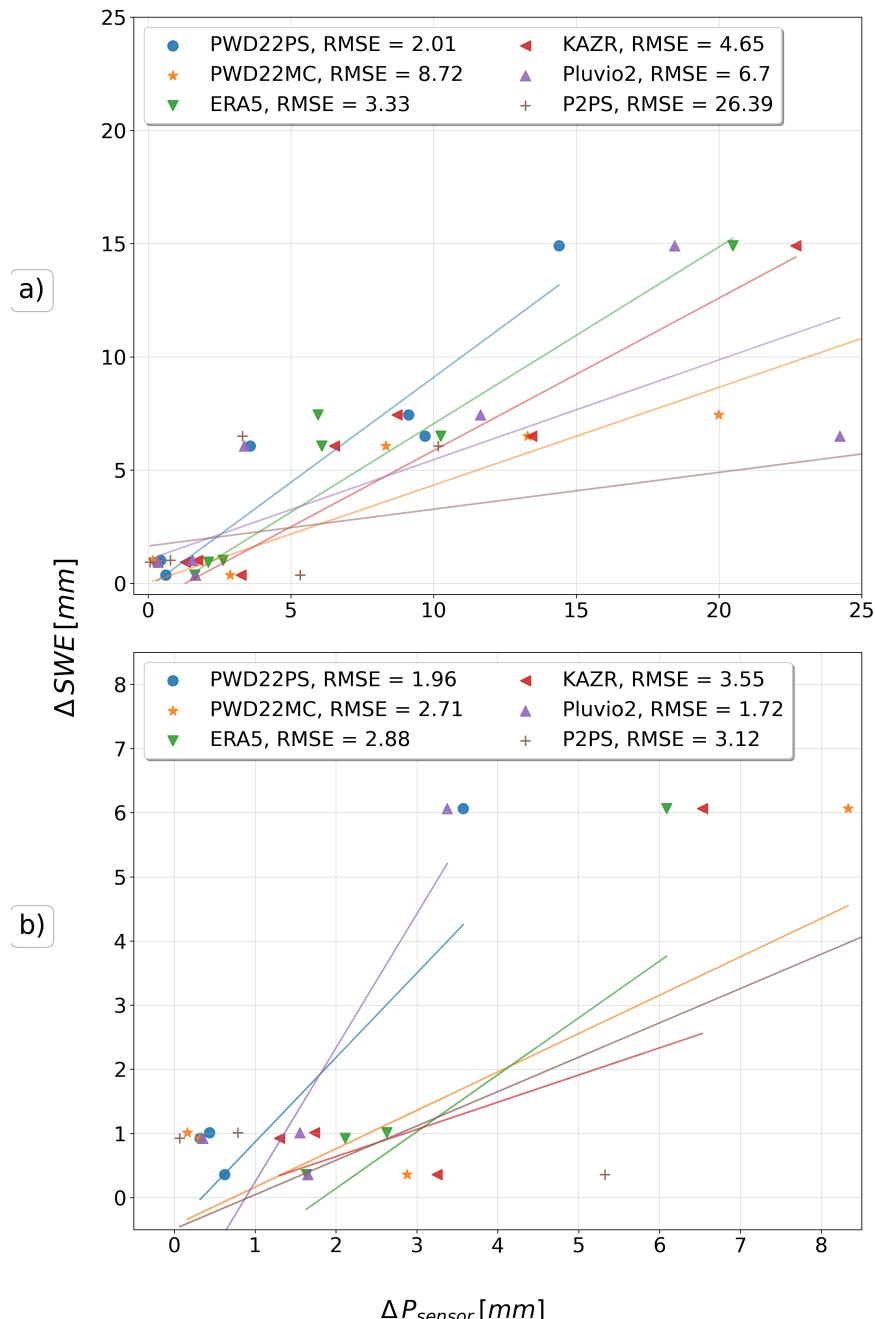

**Figure 12.** a) RMSE [mm] of the sensors and ERA5 with respect to snow cover SWE, for the time period before 20 February 2020 with n = 9 days of sampling, including days when drifting snow was detected. b) as a) but without days when drifting snow was detected before (n = 6 days). The lines show the linear regression for each sensor.

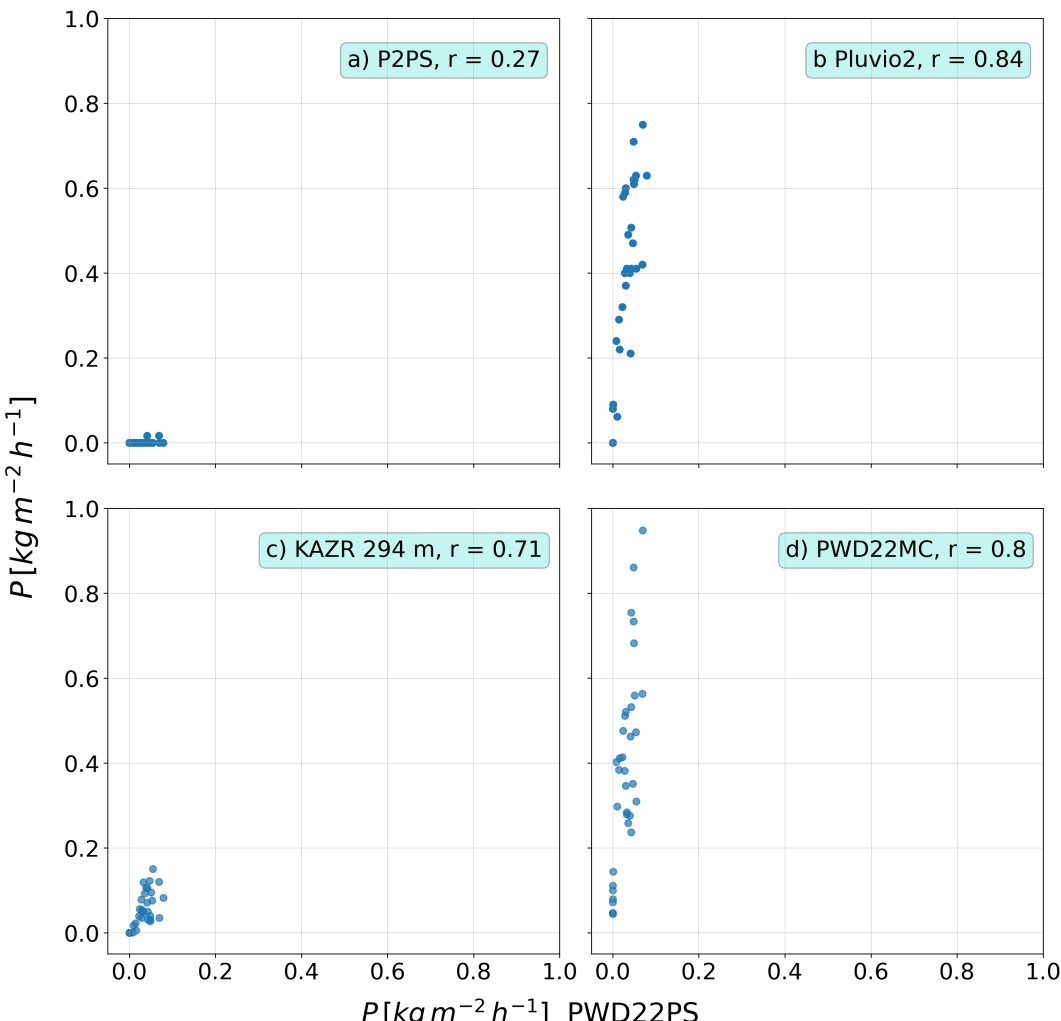

**Figure 13.** Scatter plots of PWD22PS snowfall rates vs. different sensor snowfall rates for the drifting snow event on 24 - 25 February 2020 for a) P2PS, b) Pluvio2, c) KAZR and d) PWD22MC.

during MOSAIC, SYI and FYI - why we decided to use the SWE average of both, northern- and southern loop as a reference

for snowfall sensor and reanalysis comparison. Nonetheless, a snow height difference comparison of terrestrial laser scanning (TLS) or airborne laser scanning (ALS) based digital elevation models and transect snow depth would be desirable.

For the evaluation of the SWE in- and decrease, note that the net decrease generally includes eroded mass in addition to incoming precipitated mass, hence the eroded mass was often larger than the precipitated mass. These two quantities can only be determined separately when considering snowfall events and no drifting snow events at the same time in between days of

sampling.

The constant value of SWE from 20 February on, raises the question of a saturation mass of snow that can accumulate in this

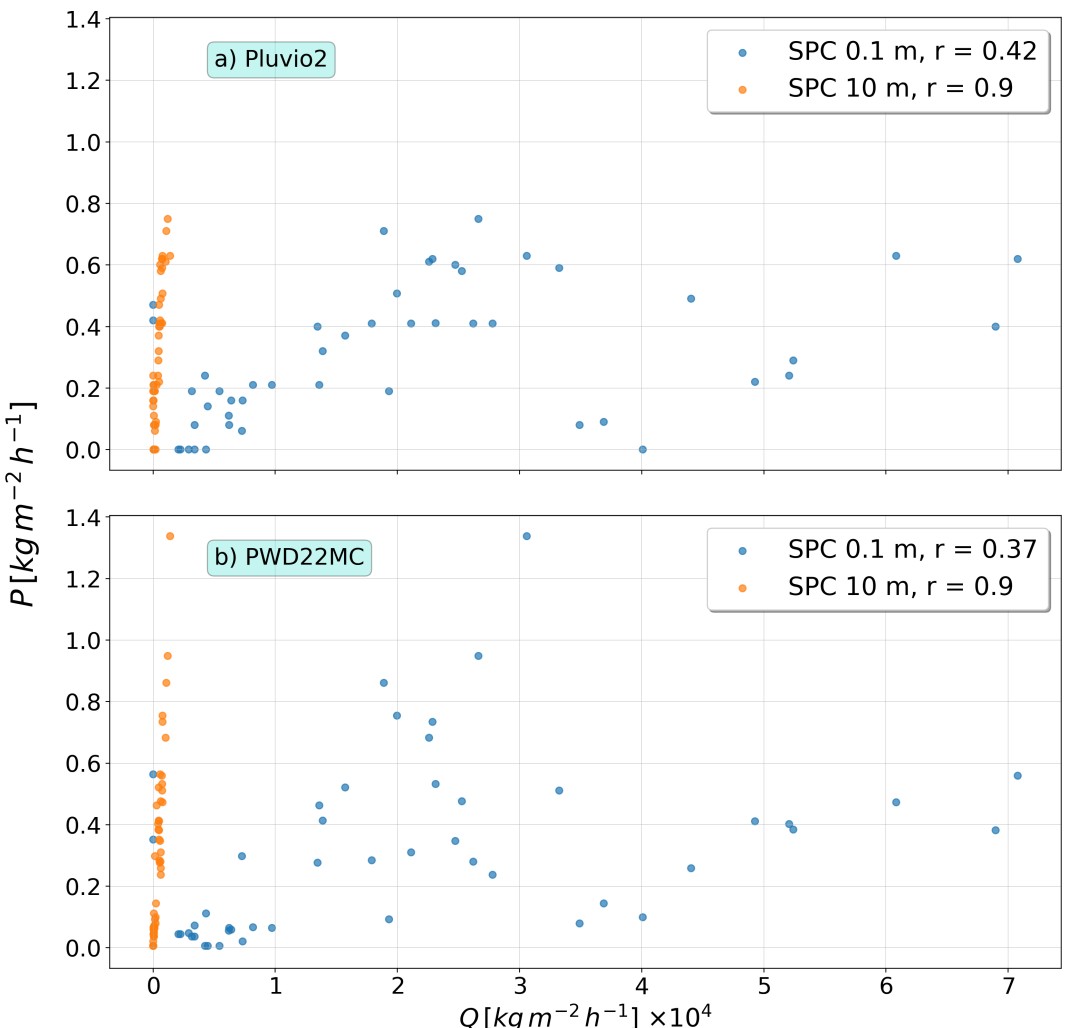

**Figure 14.** Scatter plots of sensor snowfall rates (y-axis) vs. SPC mass flux (x-axis) for the drifting snow event on 24 - 25 February 2020 for a) Pluvio2 and b) PWD22MC.

windy environment, where the roughness of the surface might be a limiting factor. Possibly, this is due to smoothening of the initially rough sea ice surface due to deposited drifting snow. The less rough the surface, the less snow can accumulate. This may be also related to the faster increase of SWE on FYI compared to SYI, and the fact that the initial 50 % of SWE over FYI reached 90 % of the SWE over SYI by end of the investigation period. One hypothesis might be, that over SYI, the snow mass is rather saturated, as the snow had time to smooth the ridges already over one season before, while FYI might me more rough as long as fresh ridges are existent. Weiss et al. (2011) found in general higher aerodynamic roughness lengths for SYI. With higher roughness length we expect more flow separation and more accumulation. However, the study is more than 10 years old and given the fact that average ice thickness is thinning in the context of global warming, we might see less stable FYI


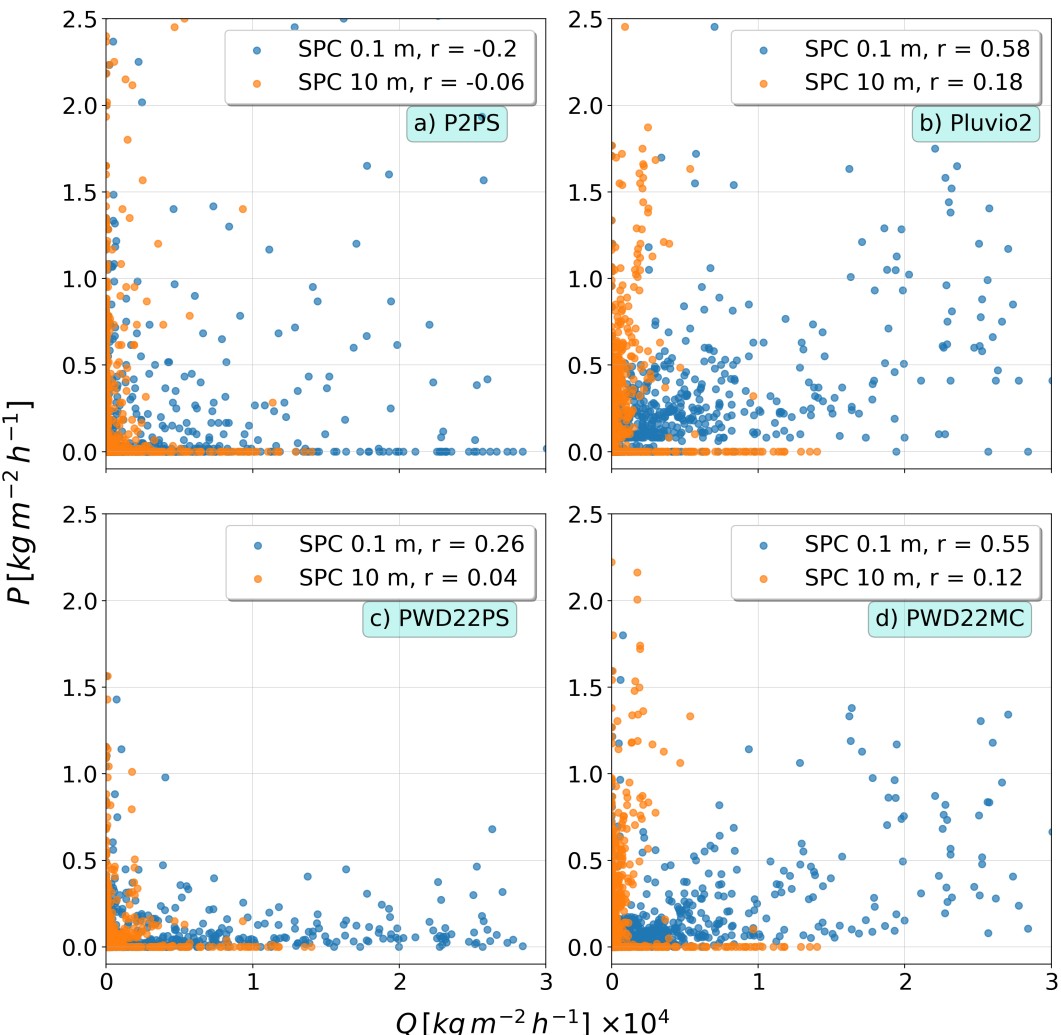

**Figure 15.** Scatter plots of sensor computed snowfall rates (y-axes) vs. SPC mass flux (x-axes) for the whole time period for a) P2PS, b) Pluvio2, c) PWD22PS and d) PWD22MC.

by the year 2019/2020, resulting in more ice motion and ridge formation - and leading to larger aerodynamic roughness lengths.

It also remains to be investigated why most drifting snow and erosion occurred from 20 February 2020 on, as indicated by transect SWE decrease and horizontal mass flux. Indeed, snow mass flux rates are high until 27 February, but on the sampling day itself, the wind speed dropped as well as the mass flux rate (Fig. 11). Between 27 Feburary and 5 March, the probabil-
ity for snow transport was very low, indicated by both, SPC mass flux and computed threshold friction velocity. One might consider that lower horizontal sampling distance ($\overline{D}_{5Mar} = 1.6\,\mathrm{m}$) could have led to an underestimation of the accumulated snow. However, from Fig. 6 one can see that reducing the sampling frequency by half has no significant effect on the average.

One can only see a significant fluctuation from reducing the frequency to 1/3 and below. Hence, a reduced sampling frequency does not explain the mass decrease. Ice dynamics that affected the transect occurred from 11/12 March 2020 and thereafter.

Hence, no ice dynamics influenced the sampling on 5 March 2020. Nevertheless, an impact on the time series cannot be excluded from 12 March 2020 on. Looking at horizontal mass flux and decreased computed snow mass together, we cannot say for sure that erosion was largely responsible for the mass decrease between 27 Feburary and 5 March. The threshold friction velocity was still exceeded on the 27 February, indicating snow transport, while low recorded mass flux indicates low transport. The computed threshold friction velocity after Bagnold (1941) (Fig. 11e) does not indicate much more or less drifting snow

compared to other time periods. However, the formula after Bagnold (1941) is rather simple and neglects varying temperatures and therefore varying bond strengths of snow particles at the top of the snowpack. The bond strength depends strongly on weather history. The preceding long period of air and snow surface temperatures often below -30 °C (Fig. 11c) could have inhibited sintering, which is strongly temperature-dependent and develops more slowly at lower snow temperatures (Colbeck et al., 1997; Colbeck, 1998; Blackford, 2007). The Bagnold formula does not consider splash entrainment (Comola et al., 2017;

Comola and Lehning, 2017) or surface roughness and atmospheric stability, which largely affects the near-surface wind field. Furthermore, the used bond strength parameter of $A = 0.18$ we used in Eq. 2.6, was found by Clifton et al. (2006) in a wind tunnel, with an ambient temperature range of -16 to 0 °C. However, this temperature range was undershot most of the time during our investigation period of MOSAiC (Fig. 11c).

In any case, we expect the transect time series until and including 5 March 2020 to be mostly valid. However, the usage of the

transect SWE for sensor- and reanalyis comparisons is limited, as erosion has very likely occurred over some periods between the transect sampling days. Hence, the amount of erosion is not quantifiable based on the available transect data. Nonetheless, days where it is likely that no erosion occurred since the previous transect sampling prior to 20 February 2020 could be detected by means of snow particle counters. The cumulative transect SWE of these time periods was then compared against cumulative SWE from the sensors and ERA5 for the same time periods, which will be discussed in the next section. The validity of the

SWE time series decreased with increasing ice dynamics from 11/12 March 2020 on and with decreasing temporal sampling frequency on both northern and southern transect loop.

If we assume, based on the findings, that PWD22PS is least affected by blowing snow, and provides a reasonable estimate of snowfall, then the snowfall between 31 October 2019 and 26 April 2020 was about 72 mm. We know that the PWD22PS showed the lowest cumulative snowfall and a systematic negative bias compared against ground truth (Fig. 12). Further, it has

been suggested that PWD22PS tends to underestimate snowfall, with values of down to 30 % less compared to a Geonor gauge within a DFIR (Wong (2012a), Tab. 1). Until 7 May 2020 as end of MOSAiC leg 3, the total snowfall would then be at least about 98 mm in 189 days. If we assume the estimated snowfall from the KAZR as upper limit for cumulative snowfall (as it was demonstrated that Pluvio2 and PWD22MC were affected by wind and blowing snow towards overestimation), we find about 114 mm. This value is comparable with the cumulative snowfall from ERA5 (110 mm). Hence, during 189 days, was we have

as the best estimate that the total snowfall was between 98 and 114 mm. With the total mass increase of 38 mm for the transect SWE during the full observation time, we can approximately compute the minimum total eroded mass as 34 mm until 26 April 2020. With almost 50 % of eroded snow mass, we find comparable magnitudes as Leonard and Maksym (2011), although they

investigated snow over Antarctic sea ice and their time period for investigation was only about 1 month. However, our findings also compare well to results from Essery et al. (1999). Further sensor assessments discussions are made in the next section.

Sublimation of snow crystals during trace precipitation or diamond dust could have led to snowfall detection in the optical sensors but no SWE increase of the snow cover. However, the study design does currently not allow to investigate that but it is recommendable to be investigated in detail. Earlier studies suggest that blowing snow sublimation may be responsible for about 6 % for the mass sink (Chung et al., 2011). The relatively low sublimation rates also arise from the high relative humidities found over sea ice, inhibiting further sublimation due to a quick saturation of the air. Snow cover sublimation can generally be

expected to be negligible during polar night (Webster et al., 2021).

Studies of Déry and Tremblay (2004), Leonard et al. (2008) and Leonard and Maksym (2011) indicate that, besides sublimation, large parts of drifting and blowing snow will drift into the open water of leads. However, for the drifting snow event on 24 - 25 February, there was no significant open water area in the vicinity. Thus, without such a local sink, we would expect that even though snow was eroded, more snow-mass would be delivered from the upwind side at approximately the same

amount. However, the given mass of drifting snow from the upwind side depends on the low-pressure system's extent and trajectory associated with the high wind speeds. In Fig. 11 we see that after the event, some atmospheric parameters have changed significantly and rapidly (a temperature drop, a decrease in wind speed). The rapid change of atmospheric conditions could explain why no "new" snow has been transported from the windward side. Another important factor could be, that the area that the transect covers, has a relatively low surface roughness, with ridges that were generally shorter than in the sur-

rounding region. Erosion might be large over these relatively smooth areas that the transect covers, and the delivered drifting snow from the upwind side could have been deposited at higher ridges upwind. But even a just frozen lead will be filled by drifting snow after a while, that would lead to a mass sink on the upwind side, inhibiting further snow transport downwind. Considering the ALS-based digital elevation map of the floe (Fig. 2) and the wind direction around 24 - 25 February (between North and North-East) (Fig. 11b), it is evident that the northern transect was likely to be partially wind-shadowed by higher

surface structures on the upwind-side. This wind-shadowing can lead to less wind transport downwind but maybe also to less erosion due to lower surface friction. This is suggested by our findings, too, as we found more substantial erosion over the northern transect loop. During the drifting snow event, the southern transect was often in the wind-shadow of the vessel and other installations (Fig. 2), leading to deceleration and less surface friction that could have led to snow transport or erosion. In the case of more deposition upwind, the mass over the sampled area would decrease even if the friction velocity behind the

obstacles is decreased, which leads to less erosion. The computed SWE would not be representative for the whole floe in this case. While potentially interesting, the method used in our study does not allow to investigate these balances. Another reason for a computed SWE decrease during the later part of the observations might be wind-induced compaction of the upper snowpack. With our approach, we assume almost a linear relationship between snow depth and SWE. However, snowdrifts consist of dense packed, rounded grains (Fierz et al., 2008). Hence, in this case, we probably under-estimate the increase (decrease)

of the SWE in case of drifting snow deposition (erosion). A thorough investigation is recommended, but beyond the scope of this study. Further studies based on SMP measurements are planned. Another reason for underestimating snow depth with spatio-temporal sampling approaches might be that parts of the snow are caught in the porous parts of ridges, depending on

wind direction and on the age of the ridges. This is something that is not covered by this study. Furthermore, for the erosion, the upper snowpack's microstructural composition plays a role, especially the strength of the bonds associated with sintering
and temperature.

## 4.2 Snowfall sensor estimates

Snowfall rates from precipitation gauges show large differences among each other. Battaglia et al. (2010) already showed that OTT Parsivel instruments overestimate the number of large particles. Further, the uncertainty they found regarding fall velocity
was high. Wong (2012a) also found a large overestimation of snowfall for the OTT Parsivel of about 50% (Tab. 1), while the overestimation became larger during high wind speeds. The relatively low accuracy of $\pm 20\%$ for snowfall as given by the manufacturer compared to other instruments does make strong overestimations not surprising. Observations examined here appear to confirm this overestimation. Although the installation on the top deck of RV Polarstern at 24 m height appears to be well protected against blowing snow (Fig. 15a) the overestimation of P2PS rather appears to be due to wind itself as also found
in earlier studies.

Regarding Vaisala PWD22 and snowfall, Boudala et al. (2016) found that in comparison with manual measurements, it overestimated snowfall by about 33 % due to detected snow crystals not observed by the human observer. Wong et al. (2012b) instead, found good agreements with two Vaisala VRG 101 (double Alter shielded) and two OTT Pluvio (Tretyakov shielded)
gauges. Compared to a Geonor gauge in a Double Fence Intercomparison Reference (DFIR), Wong (2012a) found for the PWD22 an underestimation for solid precipitation of 32 %. Hence, we expect and also can confirm an underestimation of snowfall during MOSAiC. However, the RMSE was with 1.1 mm the lowest compared against all other validated sensors. Wong (2012a) found little wind influence on the PWD22 measurements, which we can confirm, as the RMSE relative to SWE estimates was reduced least (as relative value) compared against all other sensors after drifting snow periods were eliminated
from the comparison. Compared against SPC mass flux, we demonstrated that the installation on the railing on the top deck of RV Polarstern has protected the sensor from blowing snow. On the contrary, the same device installed in Met City on the ice showed a relative overestimate, suggesting an influence from blowing snow. Comparisons with the KAZR-derived snowfall, which is also unaffected by blowing snow, further support the notion that the PWD22PS was not significantly impacted by blowing snow.


The undercatch of snowfall due to the wind for weighing bucket gauges is well known (Goodison et al., 1998; Boudala et al., 2016; Kochendorfer et al., 2017). However, we observed a strong positive bias in the cumulative snow mass for the Pluvio2 gauge when comparing with PWD22PS. This bias was largely due to a strong drifting snow event at the end of February, probably leading to blowing snow being lifted from the ground and landing in the bucket (Fig. 14,15). Available transfer functions
(e.g., Goodison et al. (1998); Boudala et al. (2016); Kochendorfer et al. (2017, 2018)) cannot correct for this type of blowing snow event, as they correct the underestimation with increasing wind speed. However, we suggest that these events can be

detected using snow particle counters and removed before applying other corrections.

The snowfall retrieval from the KAZR using a $Z_e$-S relationship with coefficients determined by Matrosov (2007) shows an overestimation relative to the snow cover SWE but performs comparably well as ERA5. The difference to ERA5 in cumulative snowfall was only about 5 mm by end of the investigation period. Considering the known low bias in PWD22 measurements, and the fact that the radar snowfall retrievals are not affected by wind or blowing snow, it is likely that this radar-based estimate provides reasonable results. However, such radar-based retrievals are dependent on the inherent properties of the snowfall observed in the datasets from which they are derived.

## 4.3 ERA5 performance

ERA5 performed reasonably well, with a weak overestimation when compared with snow cover SWE. Overall, the timing of the snowfall events is represented well by ERA5. Relative to PWD22PS, the cumulative ERA5 snowfall would have been overestimated by about 25 % by 7 May 2020. However, as already discussed, due to the light underestimation of the PWD22PS, the true snowfall lies probably between PWD22PS and ERA5 or the KAZR. Relative to PWD22PS, ERA5 appears to perform better before about March 2020, although the available data does not allow to prove that it generally performs worse before. As the ERA5 performance depends on input of measurements and numerical weather data, we can at least point out that there was a substantial decrease of the air- (minus 50 - 75 % between March and May 2020), che) and ship based observations which may lead to a worsening of the performance. che found globally a worsening of temperature forecasts of up to 2 °C in the time period March - May 2020, compared against February 2020. Nonetheless, a comparison with the cumulative sum of the KAZR (Fig. 10b) rather speaks against this theory as the ERA5 cumulative snowfall is always below the KAZR snowfall. Exact reasons with respect to ERA5 should be thoroughly investigated. Cabaj et al. (2020) also found a general positive bias (in daily snowfall rates) comparing ERA5 with CloudSat data. However, CloudSat is only available up to 82 °N, while MOSAiC was most of the time north of 84 °N and for large parts even above 87 °N (Fig. 1) between October 2019 and May 2020. Further, in Cabaj et al. (2020), they found a decrease in the positive bias towards the spring months compared with the winter months of December, January, and February while we find an increased apparent bias during this time when comparing with PWD22PS. Wang et al. (2019a) compared ERA5 snowfall with data from several snow buoys for a comparable drift track in the Arctic ocean to MOSAiC and a comparable time spans as our validation, from autumn to spring in the following year. Although they do not give details about the method, they mention a positive bias of ERA5 cumulative snowfall of about 63 mm compared with snow buoys by end of the investigation period. They found partially a negative bias and very varying results but on average a positive bias. However, as already pointed out, it cannot be concluded from a few point measurements on the overall accumulated snow on a larger area. Further, from our study we see that we cannot conclude from the computed SWE of the snow cover alone on the precipitated sum as the erosion appears to be large, even when measured over larger areas. Nonetheless, the results from Wang et al. (2019a) also indicate that a lot snow mass gets eroded over time, which is even more evident as we as well as Cabaj et al. (2020) find an overestimation tendency for ERA5 snowfall. We must further notice that ERA5 does not consider blowing snow sublimation in their computations. Although Orsolini et al. (2019) do not find a

mass effect of including blowing snow sublimation in ERA5, Chung et al. (2011) for instance, computed a larger effect of blowing snow sublimation of $12\,\mathrm{mm\,y^{-1}}$ over sea ice over 324 days. The non-existent blowing snow particle sublimation may be a reason for the overestimation tendency found in our study. Recent results with a new model of drifting snow sublimation (CRYOWRF- (Sharma et al., 2021)), indicate that it may be more important than previously estimated. We plan to address this particular problem in our future work.

## 5   Conclusions

We fitted a snow depth-SWE function to computed SWE values from SnowMicroPen force signals and applied the model to snow depths from Magnaprobes for the northern- and southern transect loops of MOSAiC, for the winter and spring time period between October 2019 and May 2020. We show that the SWE reconstructions compared well against direct SWE measurements. Besides transect paths, other snow depth sources, such as snow height differences from terrestrial laser scans, could be used with this function to compute SWE.

One particular finding of our work is that SWE differences between snow on deformed SYI and snow on remnant SYI and FYI decrease until the end of the snow accumulation season at the beginning of May. The SWE on remnant SYI and FYI, while starting out at only about 50 % in late October 2019, reached almost 90 % of the value for snow on deformed SYI by end of April. Since SWE also did not increase much after 20 February, the observations raise the question as to whether there is a saturation point for snow mass accumulation. We suggest a range of 34 mm (47 %) to 69 mm (68 %) of precipitation that has been eroded over time, with the PWD22PS lower and the KAZR as upper limit of cumulative snowfall. There was a remarkable snow erosion event between 24 to 25 February 2020, which we decided to have a closer look onto as snow cover SWE decreased during this time. The fate of the eroded snow is unclear, but it is likely that a significant amount was deposited around higher ridges or filled in the gaps of frozen leads, limiting its transport to the areas covered by the transects. However, as transects were conducted approximately weekly, processes in the snow that have occurred in the meantime are not detected. More snow erosion accompanies the temporal adjustment of the surface roughness between remnant SYI and deformed SYI. Besides temporal sampling frequency of the transect, another limitation of the study is that layering of the snowpack was not considered for estimating deposited and eroded snow mass. However, validation measurements at different points in time suggest that the impact of this effect might be small. A thorough investigation of layer density is recommended. Further research, connecting snow microstructure with snow transport rates, for instance, investigating sintering and bond strengths of snow grains that depend on temperature, could help to elucidate high Arctic snow processes. This is also important as the eroded snow has influenced several sensors' measurements.

Although the unquantified eroded snow mass limits the potential for inter-comparisons, we found that the Present Weather Detector 22 optical precipitation sensor installed on Polarstern showed the smallest differences with the estimated SWE during periods without indication of drifting snow. We demonstrated that this PWD22 was mostly protected from blowing snow influ-

ences while this blowing snow led to high correlations between horizontal mass flux and precipitation rates of the pluviometer and the Present Weather Detector at Met City. We assume that the high snowfall rates detected with the ice-based pluviometer originates to significant parts from blowing snow that was blown into the gauge opening. Similar to PWD22 on Polarstern, the KAZR snowfall retrievals are likely not affected by wind or blowing snow. The snowfall retrieved with this $K_a$-band radar shows a relatively high bias relative to the snow cover SWE, and it appears to overestimate snowfall. This overestimation however can be reduced when doing more research on selecting range gates and fitting of the coefficients. The OTT Parsivel[2] laser disdrometer showed a strong tendency towards overestimation. For the Present Weather Detector 22, installed on Polarstern, we find a total cumulative snowfall of 98 mm over 189 days. We suggest the KAZR snowfall as possible upper cumulative limit with 114 mm. Snowfall rates from the ERA5 reanalysis showed a reasonable performance, with a good timing of snowfall events but a relatively safe tendency towards overestimation. This light overestimation may arise from non-existing blowing snow sublimation. However, to our knowledge, we present the first validation of ERA5 snowfall for the high Arctic, based on a combination of repeating snow depth transect and a set of in-situ precipitation sensor data.

This study sets the base for future snow mass balance research for MOSAiC and further general snow research for Arctic sea ice. We are better aware of snow mass quantities that accumulate and erode over the central Arctic sea ice surface for almost the whole accumulation season, including the polar night. The data can be used to improve numerical snow cover-, sea-ice, weather- and climate models and for more detailed process understanding research across disciplines. Initial sensor validations were conducted, which allow for more specified, more thorough research.

*Data availability.* KAZR- (Lindenmaier et al., 2019), PWD22- (Kyrouac and Holdridge, 2019), Parsivel[2]- (Shi, 2019) and Pluvio[2] data (Wang et al., 2019b) are publicly available in the ARM data archive. Derived SWE from the SnowMicroPen is available on PANGAEA (Wagner et al., 2021). Raw SMP data is available on PANGAEA (Macfarlane et al., 2022a). Bulk SWE measurements are available on PANGAEA (Macfarlane et al., 2022b). Magnaprobe snow depths are available on PANGAEA (Itkin et al., 2021). Raw met tower data is available on the Arctic Data Center (Cox et al., 2021). SPC data will soon be available on the UK Polar data centre (https://www.bas.ac.uk/data/uk-pdc/).

## Appendix A: SWE comparison with Sturm et al. (2002a)

We can compare the HS-SWE function ($SWE = 323.97 \cdot HS^{1.07}$, with HS in meter) with the fitted function from Sturm et al. (2002a) ($SWE = 0.348 \cdot HS$, with HS in centimeter), derived for sea ice snow cover during the SHEBA campaign (Perovich et al., 1999; Uttal et al., 2002). It is shown that the fit in this paper (hereby called "W2021") generally delivers lower SWE values for the same snow height compared against Sturm et al. (2002a) (hereby called "S2002"). The difference becomes larger the higher the snow depth is, which is partially a result of the different gradient (323.97 vs. 348). However, there is also a faster change between 0 and 200 mm, which is a result of the slight non-linearity of the W2021 function. The overall difference is significant, as the deviation of the overall average from S2002 (86.5 mm) to W2021 (74 mm) is +17 %, while the RMSE is

13 mm. This deviation is relevant, especially when comparing ground truth with snowfall sensors in such a dry environment

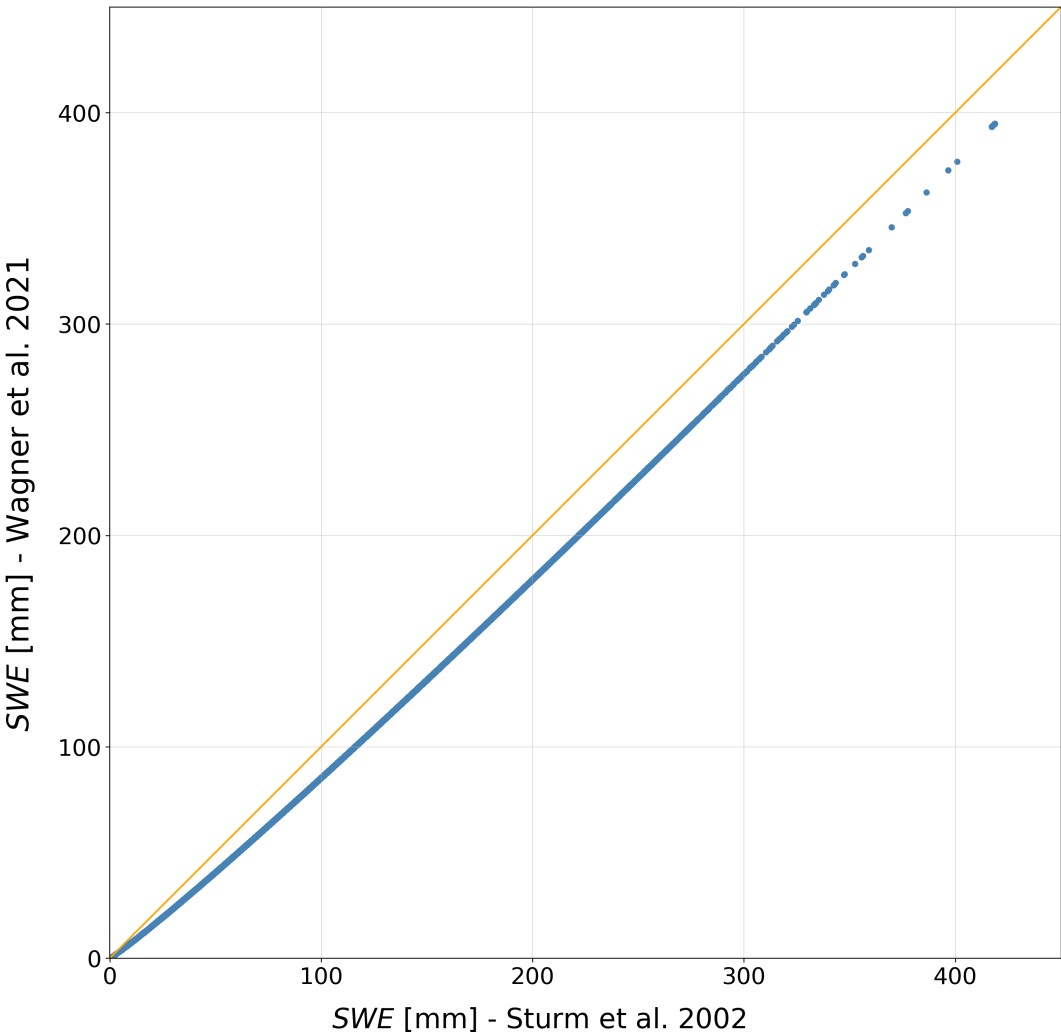

**Figure A1.** Scatter plot of derived SWE after fitted function in this paper and function by Sturm et al. (2002a). The orange line marks where r = 1.0.

as the Arctic, where average yearly snowfall rates are typically very small. These points justify the application of a specific fit for the winter and early spring during MOSAiC, instead of using values from literature. Differences in comparison with Sturm et al. (2002a) stand out very quickly: They found an average snow depth of 0.337 m and an average density of 340 kg m$^{-2}$ resulting in 166 mm SWE. We found and average snow depth of 0.249 m and an average density of 286.5 kg m$^{-2}$. Hence, both values are lower in our study resulting in an average SWE of 71.2 mm. This is indeed less than the half found by Sturm et al. (2002a). Note that the timing and setup of the snow measurements of Sturm et al. (2002a) are indeed comparable to ours, as they conducted snow transect measurements along transects for the accumulation period October to May, as well. However,

885

they used different instrumentation and also sampled over different horizontal extents. As discussed above, the average SWE change due to a snow transport event can be very different when we compare one transect section SWE against the whole transect SWE. Hence, comparability is always subject to great uncertainties. Also note, that Sturm et al. (2002a) found a 25 % higher value of snow cover SWE, compared against snowfall measurements with a Nipher shielded gauge, even after applying a 2.5 factor as precipitation correction against undercatch. The lower SWE on the snow cover is in any case very contrary to our results, as we estimate at least 56 % less snow mass on the ice compared against measured snowfall. A larger impact on the results could be due to the fact, that the measuring during our investigation period did mostly occur between the 85th and 83th latitude, while during SHEBA, the camp started at around 75 °N, 142 °W and drifted up to 75 °N, 162 °W. We can speculate that a less dry climate further away from the pole led to more snowfall, or that certain weather patterns led to notable differences over longer time spans in snowfall, wind conditions, or temperature, which we cannot determine at this point. Given that fitting the HS-SWE function is highly specific for given snow conditions and considering the substantial different average snow depth and snow density, we conclude that these are the main reasons for different fitting parameters that are found for our study. Nonetheless, the measuring setup and instrumentation itself might lead to differences, and cannot easily quantitatively compared.

*Author contributions.* DNW took the lead in writing the manuscript and implemented comments, improvements and findings from discussions with co-authors. DNW, JR, NK, ES, MO, IR, MS, MJ, DK, ARM, and SA conducted the snowpit measurements. DNW and ARM processed the snowpit data. The principal investigator for snow pits is MS. SH, MO, PI, MJ, SA, JS and RR conducted transect measurements. The principal investigator for the transect is PI. Transect data was post-processed by DNW. MDS, OGP, TU, MMF, AK and CC were responsible for the met tower measurements, including SPC. The principal investigator for the tower is MDS. The University of Colorado / NOAA flux team processed the met tower data. The principal investigator for the SPC is MMF. DNW post-processed the SPC data. Team ICE created floe maps during the cruise. SH provided the code to correct the coordinate transformation from global to FloeNavi coordinates. The KAZR and all precipitation sensors were set up, maintained, and processed by the ARM team, and MDS was the principal investigator for the ARM involvement in MOSAiC. DNW conducted further KAZR and gauges processing. MN was leading the ice team during MOSAiC and was coordinating the work of the different sub-groups within the ice team. ML was involved in MOSAiC planning and supervised the work during the manuscript.

*Competing interests.* The authors declare that they have no conflict of interest.

*Acknowledgements.* Data used in this manuscript was produced as part of the international Multidisciplinary drifting Observatory for the Study of the Arctic Climate (MOSAiC) with the tag MOSAIC20192020 and the project ID: AWI_PS122_00. DNW and ML were supported by the Swiss National Science Foundation (SNSF-200020_179130). KAZR reflectivities and precipitation data were obtained from the Atmospheric Radiation Measurement (ARM) user facility, a U.S. Department of Energy (DOE) Office of Science user facility managed by

the Biological and Environmental Research Program. "Met tower data was supported by the National Science Foundation OPP-1724551 and by NOAA's Physical Sciences Laboratory (PSL) and Global Ocean Monitoring and Observing Program (GOMO)". MDS was supported by the DOE Atmospheric System Research Program (DE-SC0019251, DE-SC0021341). PI was supported by the National Science Foundation NSF-1820927 and by the Research Council of Norway RCN-287871. SA, SH, DK, MN, RR, JR were supported through the Alfred-Wegener-Institute's funds AWI_SNOW, AWI_ICE and AWI_ROV. SA and MN were supported through the German Research Foundation (DFG) projects SnowCast (AR1236/1) and SCASI (NI1095/5). We thank Sergey Matrosov for valuable discussions about snowfall and for helping with his radar expertise. The authors thank all MOSAiC participants for contributing to the paper in one way or another: the MOSAiC logistics team, the ship's personnel, all the different team members (ICE, ATMOS, BGC, ECO, OCEAN), the ARM crew, the project coordination, and the MOSAiC project board.

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
