# Peer review of "Snowfall and snow accumulation during the MOSAiC winter and spring season"

_The Cryosphere, 2021_

## Referee Comment (RC1)

**Wagner et. al., 'Snowfall and snow accumulation processes during the MOSAiC winter and spring season'**

The authors introduce novel work completed as part of MOASiC to constrain snow mass balance and relate measured SWE (on the ice) to precipitation estimates from sensors and reanalysis on sea ice. A detailed introduction is provided defining contributions to snow mass balance along with examples of previous efforts to constrain these terms in terrestrial and sea ice domains. Strong covariance between snow depth and water equivalent is leveraged to characterize spatiotemporal change in SWE along two transects on second year ice. The depth-to-SWE relationship is built from a combination of traditional corer and SnowMicroPen estimates of bulk SWE. Blowing snow events, as indicated from particle counters, and derived horizontal mass flux, are used to delineate accumulating from drifting snow in the transect SWE data and identify periods for process attribution. Cumulative precipitation estimates from a range of sensors are compared against the transect estimates of SWE to estimate retention on the ice against erosion terms. It is demonstrated that between 54 and 68% of precipitation is lost to erosion in terms of cumulative SWE.

The introduced dataset is one of a kind and provides a critical reference for a broad range of Arctic science. I would like first to congratulate the authors on this ambitious work which will be of benefit to many research communities. I have identified areas where I felt strongly that clarification was needed to validate the measured SWE dataset as a reference. Overall, the work described in this paper adds novel information to inform our understanding of the snow mass balance on sea ice. I hope my comments are not misinterpreted as I view the underlying work as a good contribution, that with refinement, would see uptake. Thanks to the authors for their efforts to bring novel snow on sea ice and precipitation information to Arctic science.

Josh

**Reference Dataset Methods**: I struggled at times to keep track of the reference dataset methods, where for example, modifications were made within the results section to the accumulated SWE data (ie. P25 L524; P25 L532). There were also several datasets elements that appear to be strong support for the reference validation which are introduced but not used (ie. SMP measurements in known drift locations P9 L193, P10 L233 to delineate areas where the HS-SWE regression might fail; Co-located snow pits with SMP profiles P7 L190 that could be used for validation). My general feeling after several reads was that consolidation of the methods along with tabular descriptions of final reference data (number of observations + descriptive statistics + dates) would provide support for the precipitation analysis and improve flow.

**HS-SWE Conversion and Evaluation:** The conversion of snow height to SWE follows an assumption that covariance with bulk density is generally weak as compared to height. To exploit this the regression in Eqn. 6 show mixes coefficients from two sperate fits (SMP and corer derived). Given that this is an atypical statistical approach I would like to see a clear justification for the use of an arbitrary function (ie not fit to any specific dataset) instead of a validated fit from a combined SMP + ETH height/SWE dataset. Additionally, I was a bit surprised that previous work construct HS-SWE relationships on Arctic sea ice were not contrast against the methods introduced here (ie. SWE = 0.348 * hs; in CM units from SHEBA doi:10.1029/2000JC000400). Using the heights from the public SMP derived dataset from this paper,

general agreement can be demonstrated between the SHEBA experiment derived conversion and the one introduced here. Evaluating against the ETH + SMP derived dataset would be of interest. This would particularly interesting given that the ETH + SMP dataset covers an entire accumulation season, suggesting seasonal and spatial bulk density are well constrained.

[Figure]

*Figure 1: SWE as derived from Eqn 6 in this paper and Sturm et al 2002 based on the SMP height data. Red line indicates 1:1.*

Additionally, an improved explanation of how RMSE was used to determine applicability of the SMP coefficients would be beneficial for the discussion of retrieval skill. RMSE as shown in figure 4 and described in text (P11 L288) appears to describe residuals related to Eqn. 6 rather than retrieval skill associated with a specific set of coefficients. A straightforward opportunity for commenting on skill would be comparison against the collocated snow pits where 5 SMP profiles are noted as available at each (P7 L190). Without comparison against a known reference (ie snow pits) it should be made clear the RMSE characterizes the fit to the regression and not explicitly skill of the SMP derived density.

**Comparison with precipitation information:** Temporal reduction of the observed SWE dataset to only dates where both loops were available is justified as improving how representative the dataset is. I would like to see an improved example to support this. At times the comprehensive datasets introduced as distilled to very few aggregate comparisons (ie P25 L529 n = 9) but no discussion is given on the impact of limiting the temporal steps.

Specific Comments

P2 L29: Can you define what small and large-scale are in terms of area or length scale? Doing so will would help to frame the scales for the analysis and anticipated processes.

P2 L53: Similar to my last point, a definition for what a 'larger area' would be helpful to frame this.

P3 L127: 'snow pack' and 'snowpack' are both used in the manuscript.

P6 L167: Using the word direct to describe the ETH measurement might be confusing. To me, direct would imply melting down the snow and measuring it in graduated cylinder. I certainly leave this up to you, but I tend describe these in general terms as snow cores.

P6 L167: Additional information on the ETH methods would be helpful to understand the precision/accuracy of the bulk SWE estimates. For example, what type of scale was used (& with what uncertainty), what were the uncertainties associated sample capture (was it hard to get full samples in shallow snow with snow ice?), were measurement replicates were made at each location (how were outliers detected). This is not something that needs extended discussion but if the cores are to be considered baseline (instead of snow pits) I would like to understand the expected accuracy.

P7 L190-194: Can a table be provided to understand the number of SMP measurements by date and site for quick reference? Is not clear in text where and when the data points come from.

P9 L215: Are these measurements simply paced out or was there was distance reference used? Average distance between measurements here is less than the accuracy of the GPS (~5 m) so it might be helpful to provide this information.

P9 L216: Can you elaborate on what 'tip sinking into the ice' means? Mangaprobe users often describe the issue of 'overprobe' but in terrestrial domains this is unwanted penetration into the soil or vegetation. What is the interface that is being penetrated? How was it decided that a 1 cm correction was appropriate for all measurements?

P6 217: More information is needed on this error detection step for the magnaprobe. The magnaprobe is generally known for good repeatability (mm-scale). What is the justification for the 3-sigma approach? Given that snow depth on sea ice distributions are generally log-normal if feel like this could lead to omission bias for lower-frequency deep snow.

P10 L231: When I read this I felt like it was saying that the SMP was easier to use than a traditional corer. The SMP is a great tool but I'd strongly argue against it being easier to use then a corer and hanging scale in harsh conditions. They seem to be complementary datasets here where both are considered as a bulk value, by increasing N and thus providing confidence in the regressed coefficients.

P10 L233: This line indicates availability of both measurements in drifts and at pits. Why is this information not leveraged to demonstrate layering as a non-issue for drifts (not saying it isn't!) or provide a well-known reference to validate against?

P11 Eqn (3). Proksch et al 2015 describes median force (F bar) within a defined window as input but it is not indicated here or in the following text.

P11 L253: See major comments. Its not clear how the mixing of coefficients from two different regressions from two different datasets is justified over fitting the full dataset.

P11 L259: See major comments. Additionally, there is context as to why the original SMP-density coefficients may not be appropriate (using SMPv3 vs SMPv4 hardware) that is not noted here. I'm concerned that RMSE as used here is not an indicator of SMP skill while the text reads as such.

P13 L267: I don't agree that its beyond the scope of the measurements to distinguish relative depth hoar and wind slab contributions to SWE. It's a demonstrated capability of the SMP. Please rephrase this to indicate that it is beyond the scope of the study. Examples of SMP based layer classification with Random Forest (10.1109/TGRS.2012.2220549) , SVM (10.5194/tc-14-4323-2020), and ML methods (10.5194/egusphere-egu21-15637)

P13 L271: Its not clear why GPS is a critical consideration here if SMP profiles are noted as located in direct proximity to the snow pits (L273). Please revise this sentence to indicate that this comparison was not completed or provide a justification for needing cm-scale precision.

P14 P287: There appears to be sufficient data to demonstrate this quantitatively. I feel strongly that analysis would be strengthened with a brief assessment of skill rather than a qualitative assessment of Figure 5.

P20 L435: Could you provide more detail on why the loops are not considered separately and are immediately averaged?  The temporal record is also greatly impacted by the averaging, does this not potentially minimize the potential comparison periods? This needs to go beyond stating that averaging the two loops is 'more representative' and demonstrate why.

P20 L441: Could the periods selected based on this criterial be highlighted on one of the time series plots. I'm finding it hard to visualize the temporal frequency and range of compared periods.

P21 L451: See major comments. Its atypical to add processing steps in the results, removed from the original dataset description. Consolidation of methods to develop the reference dataset may improve flow and reduce overall length.

P21 L459: The statement regarding 'slight' changes is quite vague. Consider removing qualitative statements and refining the description to provide clear quantitative results where the data is available to support.

P24 L497: Is it possible to frame the statistical significance for mass change in terms of % lost. What was the mean SWE in this small transect relative to the 12 mm of change? From the diagram itself it would seem new drift structures formed but its challenging to determine total loss. Histograms may also be useful here to show the change.

P34 L660: This is an important statement and one that I am a bit surprised is not in the site description. If the evaluated sites are catch for larger domains of the site this is information that can be included in the original site description to indicate the presence of significant ridges outside the observations that potentially act as catch.

P36 L767: The terminology 'snow surface roughness' has not been used before in this study. Do you mean variability in surface height of the snowpack?

---

## Author Comment (AC1)

Dear Josh,

we authors thank you a lot for the effort going through the manuscript and making lots of useful revision suggestions. Before we start answering to your comments, some explanations to method and data, partially, independently of the reviewer's comments:

1) As also updated shortly after submission to the editor, blowing snow dataset of 10 m (there was an issue with the script leading to wrong mass flux) and Magnaprobe paths were corrected. As already mentioned, both adapted processing steps do not affect the conclusions.

2) Data that has been changed due to comments from a reviewer (omitting 3-sigma filtering, removing Y-axis shift for the HS-SWE function) do not affect conclusions, as well.All in all, the total difference is only a few mm.

3) We omitted standard deviation in the revised version (and the conclusions on the roughness) as indeed we find a skewed SWE distribution rather than a Gaussian distribution – so we cannot conclude on spread and snow surface roughness.

4) A few co-authors were added.

In the following we come to the answers to your comments.

Reference Dataset Methods: I struggled at times to keep track of the reference dataset methods, where for example, modifications were made within the results section to the accumulated SWE data (ie. P25 L524; P25 L532). There were also several datasets elements that appear to be strong support for the reference validation which are introduced but not used (ie. SMP measurements in known drift locations P9 L193, P10 L233 to delineate areas where the HS-SWE regression might fail; Co-located snow pits with SMP profiles P7 L190 that could be used for validation). My general feeling after several reads was that consolidation of the methods along with tabular descriptions of final reference data (number of observations + descriptive statistics + dates) would provide support for the precipitation analysis and improve flow.

The consolidation of the methods is a good point which we tried to address in the revised version. This includes moving actual method descriptions from other sections to the method section. We specified snowfall vs snow cover evaluation method. Further, we removed repetitions and compressed and removed rather irrelevant information. We added tabular descriptions for a better overview especially for transect and the method including following validations, but refer for more details to the just finished SMP + Magnaprobe raw dataset on PANGAEA, which should include all important information. The detailed data descriptions are already accessable but data access is publicly available after MOSAiC moratorium in January 2023. Snow drifts were already covered in the transect (Fig. 5 b-c).

We clarified this in the revised manuscript. We intended a more generalized approach in this paper - the goal was to combine snow cover, snowfall and drifting snow. We can understand very well that detailed information about snow layering would be beneficial. We are currently working on a paper which builds on this generalized work described in this paper that investigates small-scale snow transport and includes SMP, density and detailed snow layer investigation. Thus, we also refrained here from going too much into detail of snow layer investigation in the revised manuscript. We hope that you have understanding for this.

HS-SWE Conversion and Evaluation: The conversion of snow height to SWE follows an assumption that covariance with bulk density is generally weak as compared to height. To exploit this the regression in Eqn. 6 show mixes coefficients from two seperate fits (SMP and corer derived). Given that this is an atypical statistical approach I would like to see a clear justification for the use of an arbitrary function (ie not fit to any specific dataset) instead of a validated fit from a combined SMP + ETH height/SWE dataset. This is a very good point – indeed we needed to revise this. The shift along the y-axis is a result of the nonlinear regression method. However, in reality, there must be SWE = 0 when HS = 0. Due to this fact, and the reason that the difference of b\_tube = 0.92 and b\_smp = 2.31 is only marginal, we simplified to SWE = m \* HS^a.

Additionally, I was a bit surprised that previous work construct HS-SWE relationships on Arctic sea ice were not contrast against the methods introduced here (ie. SWE = 0.348 \* hs; in CM units from SHEBA doi:10.1029/2000JC000400). Using the heights from the public SMP derived dataset from this paper,general agreement can be demonstrated between the SHEBA experiment derived conversion and the one introduced here. Evaluating against the ETH + SMP derived dataset would be of interest. This would particularly interesting given that the ETH + SMP dataset covers an entire accumulation season, suggesting seasonal and spatial bulk density are well constrained. This is an excellent suggestion which we have implemented a longer section discussing SHEBA/Sturm in the revised version. Indeed, not mentioning and comparing/discussing the Sturm et al. 2002 paper was not intentionally, but at least the main author mainly responsible for writing was indeed not aware of its existence. Another co-author provided this good reference, but (non-intentionally) shortly after the TC submission.

Additionally, an improved explanation of how RMSE was used to determine applicability of the SMP coefficients would be beneficial for the discussion of retrieval skill. RMSE as shown in figure 4 and described in text (P11 L288) appears to describe residuals related to Eqn. 6 rather than retrieval skill associated with a specific set of coefficients. A straightforward opportunity for commenting on skill would be comparison against the collocated snow pits where 5 SMP profiles are noted as available at each (P7 L190). Without comparison against a known reference (ie snow pits) it should be made clear the

RMSE characterizes the fit to the regression and not explicitly skill of the SMP derived density.

We see that there might occur some confusion, especially when for instance also reading the other SMP-density literature (King et al. 2020; Calonne et a., 2020; Proksch, 2015) where RMSE is typically expressed as error compared against other measuring methods (CT, gravi-/volumetric methods). Indeed the RMSE as computed by us is the error of the applied SWE-HS function residuals against the regression line. We clarified in the revised version to avoid confusion.

Comparison with precipitation information: Temporal reduction of the observed SWE dataset to only dates where both loops were available is justified as improving how representative the dataset is. I would like to see an improved example to support this. At times the comprehensive datasets introduced as distilled to very few aggregate comparisons (ie P25 L529 n = 9) but no discussion is given on the impact of limiting the temporal steps.

We refer to Section 3.2 (P25, L542) in the original manuscript, where we described the limitation of the method. We see a justification for the method when also inter-compare this RMSE reduction in between the different sensors:

"Note that due to the strong cumulative aspect (i.e. we compare snowfall that is accumulated always between the days of transect measurements), the difference is naturally reduced when reducing the sample number for RMSE. Nonetheless, the fact that there is an overall tendency towards a decrease in the apparent overestimation of the sensors relative to the SWE, indicates that erosion likely did occur before the days eliminated in Fig. 13b. For the PWD22PS, the difference compared to SWE is only reduced by about 50 %, while for the PWD22MC, for instance, the difference was reduced to about 30 % of its original value. For the Pluvio2, the difference was reduced to 26 % of its initial value. While the apparent overestimation of the sensors in Fig. 13a is likely due to erosion (and hence strongly biased), the different magnitudes of RMSE reduction (Fig. 13b, Tab. 2) suggest that PWD22PS is less affected by overestimation due to high wind speeds that accompany blowing snow, compared to PWD22MC or Pluvio2, both of which were installed near the surface."

However, this was in the results part and therefore we added it in the discussion in the revision.

**Specific Comments**

P2 L29: Can you define what small and large-scale are in terms of area or length scale? Doing so will would help to frame the scales for the analysis and anticipated processes. Revised – we defined the "small scale distribution" as "decimeter to hectometer-scale snow cover area".

P2 L53: Similar to my last point, a definition for what a 'larger area' would be helpful to frame this.

Revised – we defined as and changed to "Considering a larger area (i.e. above hectometer scale up to a scale of the whole Arctic ice pack)...".

P3 L127: 'snow pack' and 'snowpack' are both used in the manuscript. Revised - Unified to "snowpack".

P6 L167: Using the word direct to describe the ETH measurement might be confusing. To me, direct would imply melting down the snow and measuring it in graduated cylinder. I certainly leave this up to you, but I tend describe these in general terms as snow cores. Revised as "bulk SWE measurements".

P6 L167: Additional information on the ETH methods would be helpful to understand the precision/accuracy of the bulk SWE estimates. For example, what type of scale was used (& with what uncertainty), what were the uncertainties associated sample capture (was it hard to get full samples in shallow snow with snow ice?), were measurement replicates were made at each location (how were outliers detected). This is not something that needs extended discussion but if the cores are to be considered baseline (instead of snow pits) I would like to understand the expected accuracy.

We provided appropriate information incl. uncertainties, referring to evaluations by López-Moreno et al. (2020).

P7 L190-194: Can a table be provided to understand the number of SMP measurements by date and site for quick reference? Is not clear in text where and when the data points come from.

We added referring to the SMP raw dataset that is soon published on PANGAEA.

P9 L215: Are these measurements simply paced out or was there was distance reference used? Average distance between measurements here is less than the accuracy of the GPS (~5 m) so it might be helpful to provide this information.

We changed to: "Ice dynamics affected the transects especially from 11/12 March 2020 on, where leads and cracks opened throughout the paths. Overall, we tried to minimize the influence of these ice deformation events on the transect measurements. However, an impact on the time series cannot be excluded. On the transects, snow height measurements were sampled with the Magnaprobe with an average distance between measuring points of 1.1\,m. Note that this value is simply an average that contains the uncertainty of GPS localization, coordinate transformation, and the step length of the user, while the users varied mainly between each leg of MOSAiC. The average distance between transformation."

P9 L216: Can you elaborate on what 'tip sinking into the ice' means? Mangaprobe users often describe the issue of 'overprobe' but in terrestrial domains this is unwanted

penetration into the soil or vegetation. What is the interface that is being penetrated? How was it decided that a 1 cm correction was appropriate for all measurements? This was indeed estimated but as we do not have a scientific justification for this, we removed this "correction". Note that all snow depths are now 1 cm higher which also affects marginally the SWE.

P9 217: More information is needed on this error detection step for the magnaprobe. The magnaprobe is generally known for good repeatability (mm-scale). What is the justification for the 3-sigma approach? Given that snow depth on sea ice distributions are generally log-normal if feel like this could lead to omission bias for lower-frequency deep snow. It is often common to apply a x-Sigma outlier detection for several applications. However, only if we find a Gaussian distribution we can assume with 3 Sigma that we would include 99.7 % of the points. This should be sufficient but also eliminating erroneous outliers in this case. However, as we in fact rather have a skewed distribution, we eliminated this processing step for the revised manuscript. We found that it led to a marginal bias that does not affect the overall conclusions.

P10 L231: When I read this I felt like it was saying that the SMP was easier to use than a traditional corer. The SMP is a great tool but I'd strongly argue against it being easier to use then a corer and hanging scale in harsh conditions. They seem to be complementary datasets here where both are considered as a bulk value, by increasing N and thus providing confidence in the regressed coefficients.

This is absolutely true, thus we changed to "Based on how the campaign was planned, we have considerably more SMP force measurements available (N = 3007) than bulk SWE weighing measurements (N = 195)."

P10 L233: This line indicates availability of both measurements in drifts and at pits. Why is this information not leveraged to demonstrate layering as a non-issue for drifts (not saying it isn't!) or provide a well-known reference to validate against? Indeed we described in the revised version that the measurements are partially in drifts. As described under the major comments, we (as well as others working on MOSAiC snow data) are working on much more detailed snow layering and density in relation to snow transport for near-future papers.

P11 Eqn (3). Proksch et al 2015 describes median force (F bar) within a defined window as input but it is not indicated here or in the following text. Correct – revised.

P11 L253: See major comments. Its not clear how the mixing of coefficients from two different regressions from two different datasets is justified over fitting the full dataset. See above as we changed the method.

P11 L259: See major comments. Additionally, there is context as to why the original SMPdensity coefficients may not be appropriate (using SMPv3 vs SMPv4 hardware) that is not noted here. I'm concerned that RMSE as used here is not an indicator of SMP skill while the text reads as such.

Revised – we added a detailed explanation why this comparison may be restricted. We also clarified that RMSE is with respect to regression line of the HS-SWE function.

P13 L267: I don't agree that its beyond the scope of the measurements to distinguish relative depth hoar and wind slab contributions to SWE. It's a demonstrated capability of the SMP. Please rephrase this to indicate that it is beyond the scope of the study. Examples of SMP based layer classification with Random Forest

(10.1109/TGRS.2012.2220549), SVM (10.5194/tc-14-4323-2020), and ML methods (10.5194/egusphere-egu21-15637)

This is of course true, we changed it simply to "It is beyond the scope of this study to attempt an approach that distinguishes different snow layers."

P13 L271: Its not clear why GPS is a critical consideration here if SMP profiles are noted as located in direct proximity to the snow pits (L273). Please revise this sentence to indicate that this comparison was not completed or provide a justification for needing cm-scale precision.

This is relevant for comparing Magnaprobe (coordinate-transformation) with snowpits/SMP (coordinate transformation as well).

We changed to: "Note that a clear quantitative comparison is difficult, as the accuracy of GPS measurements (2 m) and the following coordinate transformation of the Magnaprobe as well as SMP coordinates do not allow for cm-scale precision."

P14 P287: There appears to be sufficient data to demonstrate this quantitatively. I feel strongly that analysis would be strengthened with a brief assessment of skill rather than a qualitative assessment of Figure 5.

In Fig.5, we compared bulk SWE / SMP / HS-SWE for flat snow cover, but as well as for snowdrift locations. For now we do not see further need to do that, but refer to ongoing studies that this will be addressed soon.

P20 L435: Could you provide more detail on why the loops are not considered separately and are immediately averaged? The temporal record is also greatly impacted by the averaging, does this not potentially minimize the potential comparison periods? This needs to go beyond stating that averaging the two loops is 'more representative' and demonstrate why.

To justify this, we must anticipate something from the results in the following sections. We added the following: "Indeed it is impossible to determine at this point with this data set we made use of whether only the SWE derived from the northern- or southern- or the average of loops, is the best choice to evaluate snowfall. A snow height difference dataset based

on laser scanners of an area that includes both, northern- and southern transects and an area beyond that, could be used to validate transect snow depth. However, we do not make use of such a dataset here. We will demonstrate in the coming sections that initial average SWE on the northern loop is about twice the value of the average SWE on the southern loop. On one hand, we see that the initial standard deviation of snow (what we consider as "roughness" of the snow surface), on the southern loop is much lower. However, the SWE increase until January 2020 is much faster on the southern loop. Hence, we see a different accumulation rate depending on whether we measure snow depth on SYI (northern loop) or FYI (southern loop). Indeed, what "most representative" means, also strongly depends on the horizontal extent of snowfall and wind patterns, i.e. the total accumulated snow mass that has fallen over a certain area but is re-deposited due to wind. This is a problem we are not able to consider in this study but can potentially be solved by computing snow mass based on the difference of airborne- or terrestrial laser scan derived heights. The reason why we decided to use an SWE average of Northern and Southern loop as reference, can be broken down to the fact that it is simply representative in terms of that the MOSAiC ice floe consisted to large parts of these two ice types. ,,

P20 L441: Could the periods selected based on this criterial be highlighted on one of the time series plots. I'm finding it hard to visualize the temporal frequency and range of compared periods.

Revised.

P21 L451: See major comments. Its atypical to add processing steps in the results, removed from the original dataset description. Consolidation of methods to develop the reference dataset may improve flow and reduce overall length. Revised – this processing step explanation was shifted to methodology section.

P21 L459: The statement regarding 'slight' changes is quite vague. Consider removing qualitative statements and refining the description to provide clear quantitative results where the data is available to support.

We revised that. The intention was not to overload the reader with repetitions and details.

P24 L497: Is it possible to frame the statistical significance for mass change in terms of % lost. What was the mean SWE in this small transect relative to the 12 mm of change? From the diagram itself it would seem new drift structures formed but its challenging to determine total loss. Histograms may also be useful here to show the change. Revised - We set it in context accordingly.

P34 L660: This is an important statement and one that I am a bit surprised is not in the site description. If the evaluated sites are catch for larger domains of the site this is information

that can be included in the original site description to indicate the presence of significant ridges outside the observations that potentially act as catch. We added this now also under site description.

P36 L767: The terminology 'snow surface roughness' has not been used before in this study. Do you mean variability in surface height of the snowpack?

The term was first introduced on P10, where we specified now: "i.e. variability of snow surface height". Nonetheless, we have taken some distance from the term, since we can no longer refer to a standard deviation either.

---

## Author Comment (AC2)

Dear reviewer,

we thank you a lot for your very valuable comments. This certainly helps to improves our manuscript. Before we start answering to your comments, some explanations to method and data, partially, independently of the reviewer's comments:

1) As also updated shortly after submission to the editor, blowing snow dataset of 10 m (there was an issue with the script leading to wrong mass flux) and Magnaprobe paths were corrected. As already mentioned, both adapted processing steps do not affect the conclusions.

2) Data that has been changed due to comments from a reviewer (omitting 3-sigma filtering, removing Y-axis shift for the HS-SWE function) do not affect conclusions, as well. All in all, the total difference is only a few mm.

3) We omitted standard deviation in the revised version (and the conclusions on the roughness) as indeed we find a skewed SWE distribution rather than a Gaussian distribution – so we cannot conclude on spread and snow surface roughness.

4) A few co-authors were added.

In the following we come to the answers to your comments.

I have the following questions/comments/concerns which should be addressed.
Length – This manuscript is made exceptionally long through the inclusion of the snowfall measurement device assessment. I completely understand why the authors felt the need to combine the two datasets (snow thickness transects and snowfall sensor assessment) but it makes the manuscript challenging to follow and keep track of all the different sensors and approaches. A quick random sample of 20 TC papers online makes this one the longest. I would suggest that the snowfall sensor assessment effort could be its own Brief Communication and use it to feed the snow transect SWE work as a separate full paper to TCD. That is a decision for the Associate Editor and the authors.
Indeed the paper got relatively long. However, the amount and the different sources of data mean that the paper has to be a bit longer - we see the combination of snow cover and snowfall in this manuscript as a key element. Nonetheless, we see some issues when reading the paper, as also suggested by the other reviewer. We came to the conclusion as well that the manuscript needed some consolidation of the method and data description that it reads with a better flow. Hence, we invested work in better readability rather than making the manuscript shorter. We moved some explanations to the method section and added a table for a better overview, among others. This is our suggestion to you and the associate editor.

Brine/salt as a term in snow mass balance equation (Eq 2) – It is well known that snow on first-year sea ice types entrains expelled liquid brine from the near-surface sea ice volume into its cover during the fall and winter seasons through vapour pressure gradients via capillary and wicking action. Some of this salt can also be entrained from frost flowers once they erode after formation. This well-known brine volume in the basal layer of the snow cover has significant implications for heat transfer between the sea ice and atmosphere and thermal conductivity of the snow, including microwave remote sensing and satellite estimates of the snow and sea ice thickness via altimetry. So, shouldn't brine in the snow on first-year sea ice be a mass balance input (term) in the equation (even though this study investigates mainly investigates SYI). I need to understand its magnitude relative to say sublimation or evaporation or diamond dust, which are mentioned throughout. I good starting point for assessing the magnitude of this quantity are the following four papers 10.1016/j.coldregions.2009.03.009 doi.org/10.1016/0165-232X(84)90034-X doi.org/10.1080/01431169208904280 10.1029/96JC03208 The latter paper by Nghiem et al., 1997 report 2-4 mm of discontinuous brine slush on the ice surface. If this is all wicked up, it may be a relatively significant input, especially if the snow is thin.

Thanks a lot for this input. We discussed this and for us the question is not easy to answer. Please also note that this paper is mainly about surface processes – however for the sake of completeness we described the mass balance terms in the introduction. Nonetheless, we adapted the introduction according to your suggestions. As written in the introduction, this slush induced by brine expulsion may re-freeze at some point to sea ice with a mass contribution on top. Therefore, we added "B" for brine as positive mass balance term in the introduction. However, "B" may transform snow into sea ice at a later point. We tried to take this into account in the revised introduction. However, due to the complexity of this question, we cannot go into more detail including elaborate research. For the future, for sure, with the data collected during MOSAiC, including snow-ice interface scans with the microCT, there will be lot of opportunities to work on this question.
We revised in the introduction:

"$dM/dt = P \pm E_s \pm E_e – E_D – R + B – I – \nabla \cdot D – L – S$,
… B is brine mass infiltration rate into the snow cover from below, …
… Considering brine infiltration B which is often accompanied by the expression of frost flowers, Nghiem et al. (1997) found in indoor experiments a 4 mm slush layer forming beneath frost flowers. However, when snow falls onto frost flowers or a layer of brine, it gets soaked by brine, transformed into slush and, when cold enough, it is often transformed quickly into snow-ice. Hence, we assume that brine only can be a positive mass term as long as a certain ambient temperature is not undercut, where the snow begins to transform into snow-ice."

Minor editorial comments

Introduction – I'm quite surprised the work of S. Savelyev et al., 2006 is not reviewed, especially since it deals with blowing/drifting snow on sea ice. doi.org/10.1002/hyp.6118
Thank you for suggesting, we were not aware of this paper. We added this in the introduction and referred to the high relative humidity measured that probably goes along with low blowing snow sublimation rates.

L193 – add an 's' to transect
Revised.

L217-19 – Why not just say standard deviation instead of saying z-score? It's a clunky statistical term that some may not be familiar with.
Revised – we removed the processing step completely – so not required anymore.

Figure 3 caption – a bit odd to start your caption with 'All used'. Why not just start with 'Magnaprobe …'? Also, near the bottom of the caption, change 'origin' to 'originate'.
Revised.

L233-34 – 'Furthermore, as we made for each snowpit ….' Reads awkward. Consider revising to 'Furthermore, for each snowpit we made at least …'.
Revised.

L352 – I would argue that snowfall is estimated as opposed to retrieved.
It is common to use the word "retrieved" or "retrieval" for computation of parameters based on remote sensing, no?

L361 – 'off' should be 'of'
Revised.

L368 – 'As we found for 280 m the highest snowfall rates …' is awkward. Please revise.
Revised.

L385 – I don't think 90 o C is correct … 90 o ?
Thank you - revised.

L416 – remove 'take' .. or add 'into'
Revised.

L451 – rounded rounded
Revised.

L463-464 – Not so sure you should mention 'saturation' here. Why not just save to

introduce in the Discussion. If you do so, take L459-63 with you.
Revised – removed in the result section (but was already in discussion).

L475 – confusions … I'd make singular
Revised.

L508 – add 'one' before another
Revised.

L516 – Starting this sentence with 'Especially' is awkward, Just start with 'The'
Revised.

Fig 11 caption – areas … isn't there just a single green shaded area?
Revised.

L549 – 'than' … not 'that'
The sentence is: "While the apparent overestimation of the sensors in Fig. 13a is likely due to erosion (and hence strongly biased), the different magnitudes of RMSE reduction (Fig. 13b, Tab. 2) suggest that PWD22PS is less affected by overestimation **due to high wind speeds that accompany blowing snow**, compared to PWD22MC or Pluvio2, both of which were installed near the surface." – So we mean "that" here.

L551 – only one 'a'
Revised.

L558 – Not sure this is a sentence
Revised – we changed to "Taken together, we detected five significant snowfall events. If we use PWD22PS as reference..."

Fig 12 caption – 'and' before 'KAZR'L560-61 – this sentence is identical with L595-6
Revised.

L599 – decreased … remove the 'd'
Revised.

L600 – 'From 20 February 2020, also the standard deviation stabilized' … is an awkward sentence, please revise.
Revised. Indeed, we

L613 – 'strengths' is spelled incorrectly
Revised.

L618-19 – Please revise … reads awkwardly
Revised.

L628-20 - Furthermore, the used bond strength parameter of A = 0.18 that we used in the computation was found by (Clifton et al., 2006) in a wind tunnel, with a temperature range of -16 to 0 â¦C which was undershot 620 most of the time during our investigation period of MOSAiC (Fig. 12c). Not sure why this isn't in Section 2.22 or 2.3.
Revised.

L635 – 'where values found of 30% less … ' is awkward, please reword
Revised.

L636 – replace 'as' with 'at' … the end of MOSAiC …
Revised.

L743 – spelling 'buoys'
Revised.

L751 – spelling 'reason'
Revised.

L795 – They
Revised.

L796 – '… first validation of the ERA5' is a bit of bold statement (I would say Wong et al., 2019 and Graham et al., 2019 evaluated it with buoy data or other validation data among others)
Revised – we changed to "However, to our knowledge, we present the first validation of ERA5 snowfall for the high Arctic, based on a combination of repeating snow depth transect and a set of in-situ precipitation sensor data."

---

## Author Response (AR2)

We as authors thank the reviewers and the editor for reading and assessing our manuscript. We would like to keep the overall structure as in the last version, however we needed to make some minor changes:

1) Due to carelessness, there was a rather large conversion error in the conversion of the mass flux from g/cm^/min. to kg/m^2/h. Therefore, we had to convert the mass flow (see Text and Figures). However, this does not change anything of the conclusions, since for our comparisons in the very first place is the relative mass flux at different time and we do not aim here to evaluate the total mass flux. Nevertheless, it was important to correct this - we apologize for this and hope that this is not too big a problem.

2) We had to add a limit to the significance of the upper SPC data because it shows unnatural spikes in some places (Section 2.6). However, we decided to use the data set as a comparison, since it shows well the magnitude of the mass flux that can be expected at that altitude, compared to the lower SPC.

3) The lower SPC was actually installed at about 0.1 m above the surface, not 0.5 m. We corrected this.

4) Corrections of minor spelling errors.

Kind regards,
On behalf of the authors,

David Wagner